# Carbonaceous aerosol composition in air masses influenced by large-scale biomass burning: a case-study in Northwestern Vietnam

Dac-Loc Nguyen[1,2,8], Hendryk Czech[1,2], Simone M. Pieber[3], Jürgen Schnelle-Kreis[1], Martin Steinbacher[3], Jürgen Orasche[1], Stephan Henne[3], Olga B. Popovicheva[4], Gülcin Abbaszade[1], Guenter Engling[5,a], Nicolas Bukowiecki[6,b], Nhat-Anh Nguyen[7], Xuan-Anh Nguyen[8], Ralf Zimmermann[1,2]

[1]Joint Mass Spectrometry Centre (JMSC), Cooperation Group "Comprehensive Molecular Analytics" (CMA), Helmholtz Zentrum München, München, 81379, Germany
[2]Joint Mass Spectrometry Centre (JMSC), Chair of Analytical Chemistry, University of Rostock, Rostock, 18059, Germany
[3]Empa, Laboratory for Air Pollution/Environmental Technology, Dübendorf, 8600, Switzerland
[4]Skobeltsyn Institute of Nuclear Physics, Moscow State University, Moscow, 119991, Russian Federation
[5]Department of Biomedical Engineering and Environmental Sciences, National Tsing Hua University, Hsinchu 30013, Taiwan
[6]Laboratory of Atmospheric Chemistry, Paul Scherrer Institute, Villigen, 5232, Switzerland
[7]Hydro-Meteorological Observation Center, Vietnam Meteorological and Hydrological Administration, Ministry of Natural Resources and Environment, Ha Noi, Vietnam
[8]Institute of Geophysics, Vietnam Academy of Science and Technology (VAST), Ha Noi, Vietnam
[a]now at: Mobile Source Laboratory Division, California Air Resource Board, El Monte, CA 91731, United States
[b]now at: Department of Environmental Sciences, University of Basel, Basel, 4056, Switzerland

*Correspondence to*: Hendryk Czech (hendryk.czech@uni-rostock.de)

**Abstract.** We investigated concentrations of organic carbon (OC), elemental carbon (EC) and a wide range of particle-bound organic compounds in daily sampled $PM_{2.5}$ at the remote Pha Din (PDI) - Global Atmosphere Watch (GAW) monitoring station in Northwestern Vietnam during an intense 3-week sampling campaign from 23$^{rd}$ March to 12$^{th}$ April 2015. The site is known to receive trans-regional air masses during large-scale biomass burning (BB) episodes. BB is a globally widespread phenomenon and BB emission characterization is of high scientific and societal relevance. Emissions composition is influenced by multiple factors (e.g., fuel and thereby vegetation-type, fuel moisture, fire temperature, available oxygen). Due to regional variations in these parameters, studies in different world regions are needed. OC composition provides valuable information regarding the health- and climate-relevant properties of PM2.5. Yet, OC composition studies from PDI are missing in the scientific literature up to date. Therefore, we quantified 51 organic compounds simultaneously by in-situ derivatization thermal desorption gas chromatography time-of-flight mass spectrometry (IDTD-GC-TOFMS). Anhydrosugars, methoxyphenols, n-alkanes, fatty acids, polycyclic aromatic hydrocarbons, oxygenated polycyclic aromatic hydrocarbons, nitrophenols as well as OC were used in a hierarchical cluster analysis highlighting distinctive patterns for periods under low, medium and high BB influence. The highest particle phase concentration of the typical primary organic aerosol (POA) and possible secondary organic aerosol (SOA) constituents, especially nitrophenols, were found on 5$^{th}$ and 6$^{th}$ April. We linked the trace gas mixing ratios of methane ($CH_4$), carbon dioxide ($CO_2$), carbon monoxide (CO) and ozone ($O_3$) to the statistical classification of BB events based on OA composition and found increased CO and $O_3$ levels during medium and high BB influence. Likewise, a backward trajectory analysis indicates different source regions for the identified periods based on the OA clusters, with cleaner air masses arriving from northeast, i.e., mainland China and Yellow sea. The more polluted periods are characterized by trajectories from southwest, with more continental recirculation of the medium cluster, and more westerly advection for the high cluster. These findings highlight that BB activities in Northern Southeast Asia significantly enhance the regional organic aerosol loading and also affect the carbonaceous $PM_{2.5}$ constituents and the trace gases in Northwestern Vietnam. The presented analysis adds valuable data on the carbonaceous and, in particular OC, chemical composition of $PM_{2.5}$ in a region of scarce data availability, and thus offers a reference dataset from South-East Asian large-scale BB for future studies, evaluation of atmospheric transport simulation models, or comparison with other world regions and BB types, such as for instance Australian Bush Fires, African Savannah Fires, or Tropical Peatland Fires.

## 1 Introduction

Biomass burning (BB) is a globally widespread phenomenon, and emissions characterization is of high scientific and societal relevance. The fires release pollutants, which are harmful for human and ecosystem health (Stott, 2000;Kanashova et al., 2018;Pardo et al., 2020;Ihantola et al., 2020) and alter the Earth's radiative balance (Che et al., 2021;Lu et al., 2015). The fires, for instance, emit substantial amounts of carbon monoxide (CO), various volatile organic compounds (VOCs) of diverse chemical reactivity and level of harmfulness, and carbonaceous particulates such as elemental carbon (EC) and primary organic

aerosol (POA) (Akagi et al., 2011;Aurell and Gullett, 2013;Popovicheva et al., 2017a). When the emissions are atmospherically oxidized, the primary constituents may also form secondary organic aerosol (SOA) and thus increase the organic aerosol (OA) loading further (Seinfeld and Pandis, 2016). Yet, the impact of various types of biomass burning on the global radiative forcing remains poorly constrained, especially in terms of the chemical composition of the emitted organic aerosol (OA) and its related light absorbing properties (Martinsson et al., 2015;Zhong et al., 2014). Fire emissions composition is influenced by multiple

factors (e.g., fuel and thereby vegetation-type, fuel moisture, fire temperature, available oxygen). Due to regional variations in these parameters, studies in different world regions are needed.

The Northern Southeast Asia region is well known for emission-intense and recurring wildfires and burning of after-harvest crop residue during the pre-monsoon season every February to April. Atmospheric aerosol concentrations and haze events typically peak during these periods and have found previously to be correlated with BB activity, which contribute considerably

to the regional aerosol loadings (Streets et al., 2003;Carmichael et al., 2003;Gautam et al., 2013;Lee et al., 2016). Open BB plumes can be transported over long distances along with the prevailing westerly winds and may also influence the large-scale atmospheric circulation in Northern Southeast Asia (Lin et al., 2013;Reid et al., 2013;Tsay et al., 2016). The widespread occurrence of episodic open BB emission in Southeast Asia results in the "river of smoke aerosols" from near source regions over northern Thailand-Laos-Vietnam and depicts the confluence of aerosol-cloud-radiation interactions prior to entering the

receptor areas of Hong Kong, southeastern Tibetan Plateau or central Taiwan (Lin et al., 2013;Yen et al., 2013;Chan, 2003;Engling et al., 2017;Nguyen et al., 2016;Lee et al., 2016). Yet, long-term monitoring on the one hand, and detailed chemical characterization of the atmosphere`s gaseous and particulates constituents on the other hand is scarce in Northern Southeast Asia. The availability of reliable scientific data and information on the chemical composition of the atmosphere is crucial for a sound assessment of air pollution sources and air quality impacts. To get a full coverage such data must be

consistent, of adequate quality, and have to be available from various locations worldwide. Filling data gaps in data scarce areas is thus a top priority.

The Pha Din Global Atmosphere Watch (GAW) regional monitoring station, PDI, is located in the north of the Indochina Peninsula. Continuous observations of aerosol optical properties and greenhouse gases (GHG) have been implemented at PDI since early 2014, and aerosol optical properties have been previously presented by Bukowiecki et al. (2019). PDI was found

to be well suited to study the recurrent large-scale fires on the Indochinese Peninsula, whose pollution plumes are frequently transported towards the site. However, the source apportionment by Bukowicki et al. (2019) was only based on light-absorbing carbon and levoglucosan concentrations without investigation of other chemical species. Generally, so far, very few studies (Nguyen et al., 2016;Popovicheva et al., 2016;Popovicheva et al., 2017b;Pham et al., 2019) analysed the organic chemical composition of aerosol samples collected in Northwestern Vietnam during BB events. We complement the analysis by

Bukowiecki et al. (2019) and present results from the chemical analysis of $PM_{2.5}$ samples (atmospheric fine particulate matter with aerodynamic diameter $\leq 2.5$ μm) and trace gas measurements collected during large-scale BB in 2015 at PDI. We discuss the daily variations of bulk carbonaceous components (OC, EC), and provide a detailed chemical analysis of organic aerosol (OA) constituents, including anhydrous sugars (AS), methoxyphenols, n-alkanes, fatty acids, polycyclic aromatic hydrocarbons (PAHs), oxygenated PAHs (o-PAHs), and nitrophenols and add valuable data to the available body of literature.

This allows for comparison of the chemical composition of large-scale BB at PDI to other types of BB around the globe in future studies. We deploy a hierarchical clustering approach to the OA constituents for source and process identification, and

complement this analysis with interpretation of hourly trace gas data (methane ($CH_4$), carbon dioxide ($CO_2$), carbon monoxide (CO), and ozone ($O_3$)) and backward trajectory analysis from atmospheric transport simulations.

## 2 Methods

### 2.1 Site description

Aside from the measurements, data were collected at the PDI site (1466 m a.s.l., 21.573°N 103.516°E), a meteorological station of the Vietnam Meteorological and Hydrological Administration (VNMHA) and a regional station of the World Meteorological Organization's (WMO) Global Atmosphere Watch (GAW) program since 2014. Aside of instrumentation to collect meteorological data, the site is equipped with continuous in-situ observations of aerosol optical properties and trace gases, as described in Bukowiecki et al. (2019). The station is located in Dien Bien Province, which is 360 km northwest of Hanoi, 200 km south of the border with China, and 120 km east of the border with Laos (Figure 1). The province covers a vast area of 9541 km², and the population is estimated to be approximately 567,000 inhabitants as of 2017 (http://www.gso.gov.vn/). The observation site is located on the top of a hill about 1 km north of the Pha Din pass. In the vicinity of the station, the national highway AH13 runs connecting Son-La city (to the southeast) and Dien-Bien-Phu city (to the west). Within 5 km from the sampling site, there are only a few ethnic H'mong households using wood log and debris for residential cooking and heating. There are no industrial facilities in vicinity of the station.

### 2.2 Sampling campaign

An intense sampling campaign was conducted from 23[rd] March to 12[th] April in 2015 to complement continuous on-line monitoring with more detailed information on the aerosol particle composition. $PM_{2.5}$ (fine inhalable particles with aerodynamic diameter < 2.5 μm) samples were collected on 47 mm quartz-fiber filters (Whatman QM/A, Piscataway, NJ, USA) by MiniVol™ TAS (Airmetrics, Eugene, OR, USA) samplers (sampling height of 2 m) with an operating flow rate at 5 L min$^{-1}$ in the meteorological garden. The sampling duration was 24 hours, starting from 8:00 a.m. local time (UTC+7), and a total of 20 filters were collected. Sampling was conducted continuously, except for 25[th] March, when a filter had to be discarded due to a battery failure.

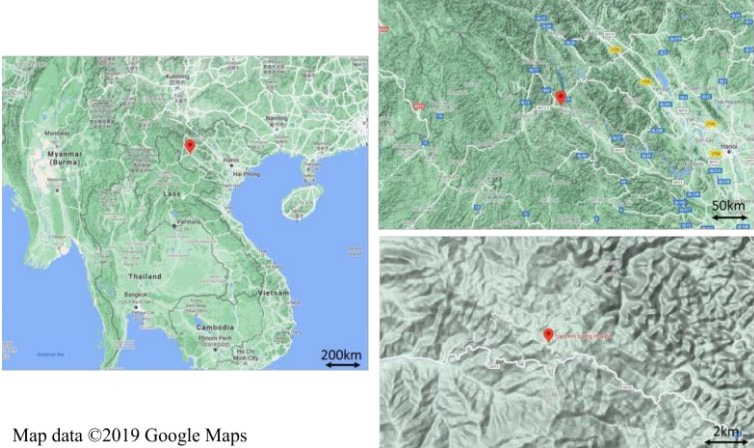

Map data ©2019 Google Maps

**Figure 1.** Maps showing PDI monitoring station, Dien-Bien Province, Vietnam, in three different scales of 200 km, 50 km, and 2 km. The maps are retrieved from the © Google Terrain Map product.

### 2.3 Analysis of organic $PM_{2.5}$ composition

Bulk analysis of organic $PM_{2.5}$ constituents was performed with a thermal-optical carbon analyzer (DRI model 2001A) for the determination of Organic Carbon (OC) and Elemental Carbon (EC) following the Improve A protocol (Chow et al., 2004a)

defining four fractions of OC (OC1-OC4) and three fractions of EC (EC1-EC3). Pyrolytic OC (OP) caused by char formation was subtracted from apparent EC1 by using the filter reflectance via a He-Ne laser at 632.8 nm.

The measurement uncertainty δy was calculated as following Equation (1):

$$\delta y = [(\text{analyte concentration} \cdot \text{instrument precision})^2 + \text{LOD}^2]^{1/2} \qquad (1)$$

with LOD being the limit of detection.

For targeted chemical analysis, the in-situ derivatization thermal desorption - gas chromatography time-of-flight mass spectrometry (IDTD-GC-TOFMS) with electron ionization (EI) was used (Orasche et al., 2011). In order to avoid thermal decomposition of analytes and lowering limits of detection, hydroxyl and carboxyl groups of compounds, for example of anhydrosugars, were derivatized by N-Methyl-N-trimethylsilyltrifluoroacetamide (MSTFA) during the step of thermal desorption from quartz fiber filter in glass-goose-neck liner at 300°C. Detected compounds were identified by library match of EI mass spectra as well as retention index and quantified by isotope-labelled internal standards of the same substance or chemically similar substance. In total, 51 particle-bound organic compounds were quantified, covering the substance classes of polycyclic aromatic hydrocarbons (PAHs), oxygenated PAHs (o-PAHs), n-alkanes, anhydrosugars (AS), high molecular weight (HMW) fatty acids (carboxylic acids with ≥20 carbon atoms), methoxyphenols, and nitrophenols.

## 2.4 Statistical data analysis

The temporal component of organic compound groups during the sampling campaign was investigated by a clustergram for visualization of sample and variable relations using Matlab® (Version 2010b, The MathWorks, MA, USA) and its Bioinformatic Toolbox. The dendrograms, which were obtained from Ward's minimum variance algorithm and Euclidean distance applied on standardized (autoscaled) variables, illustrate the similarity of variables (i.e. chemical compound class; rows) or observations (i.e. days; columns). The associated heatmap displays how many standard deviations a data point is distant from the mean. None of the included classes dropped below the limit of quantification at any sampling day, so we do not expect inflation of noise from the standardization.

A dendrogram may not accurately represent the distance matrix, which is known as ultrametric tree inequality. The cophenetic correlation coefficient denotes a metric to assess how well a dendrogram fits the distances between considered pairs of objects and is be interpreted similar to Pearson's correlation coefficient (Sokal and Rohlf, 1962).

## 2.5 Meteorological data and trace gas analysis

Records of the meteorological parameters including wind speed, wind direction, relative humidity (RH) and temperature (T) followed the guidelines of the Vietnamese National Technical Regulation on Meteorological Observations (MONRE, 2012). The records were done manually four times a day at midnight, 06.00 AM, noon and 06.00 PM UTC, corresponding to 07.00 AM, 01.00 PM, 07.00 PM and 01.00 AM local time. In addition, hourly averaged T and RH data were retrieved from ambient sensors at the main inlet for the aerosol and trace gas measurements. Precipitation was collected as daily bulk samples at the station. Carbon monoxide (CO), carbon dioxide ($CO_2$) and methane ($CH_4$) were measured with a cavity ring-down spectrometer (G2401; Picarro Inc., CA, USA); ozone was measured by UV absorption (Ozone Analyzer 49i; Thermo Scientific, CA, USA). The inlet height was 12 m above ground, roughly 6 m above the rooftop and about 200 m from the location of the MiniVol sampler, which is further detailed in (Bukowiecki et al., 2019).

## 2.6 MODIS/TERRA Fire Image and Backward Trajectory Observation Using FLEXTRA

The interpretation of the aerosol composition analyses was supported by air mass back-trajectory analysis combined with BB locations as retrieved from the Moderate Resolution Imaging Spectroradiometers (MODIS) from the Terra and Aqua platform (MODIS Collection 6 Hotspot / Active Fire Detections MCD14DL). Backward trajectories (BWT) were computed with the FLEXTRA trajectory model driven by 3-hourly meteorological analysis fields (1° x 1° resolution) of the operational Integrated

Forecast System (IFS) of the European Center for Medium-range Weather Forecast (ECMWF) (Stohl, 1996; Stohl and Seibert, 1998). Trajectories were initialized every 4 hours at different heights above the site and were followed backwards in time for 10 days within the global model domain. For the discussion, trajectories started at 420 m above ground were selected, an altitude roughly in the middle between real surface and smoothed model elevation. Fire counts were aggregated for different periods of the measurement campaign and displayed as the total number of fire counts per day on a 0.25° x 0.25° grid covering the area of interest.

## 3 Results and discussion

### 3.1 Meteorology description

Figure 2 shows the time series of meteorological observations at PDI during the sampling campaign. From 23[rd] March to 12[th] April 2015, ambient temperature ranged from 12 °C to 27°C and ambient relative humidity from 29% and 100% was observed. PDI received only occasionally precipitation with rainfall at the beginning and end of the sampling period. In the middle of the sampling period, hot and dry weather dominated with the highest observed temperature and lowest RH. Furthermore, the local wind direction was mostly from the West-Southwest during the dry period and Southeast during the middle of the sampling period.

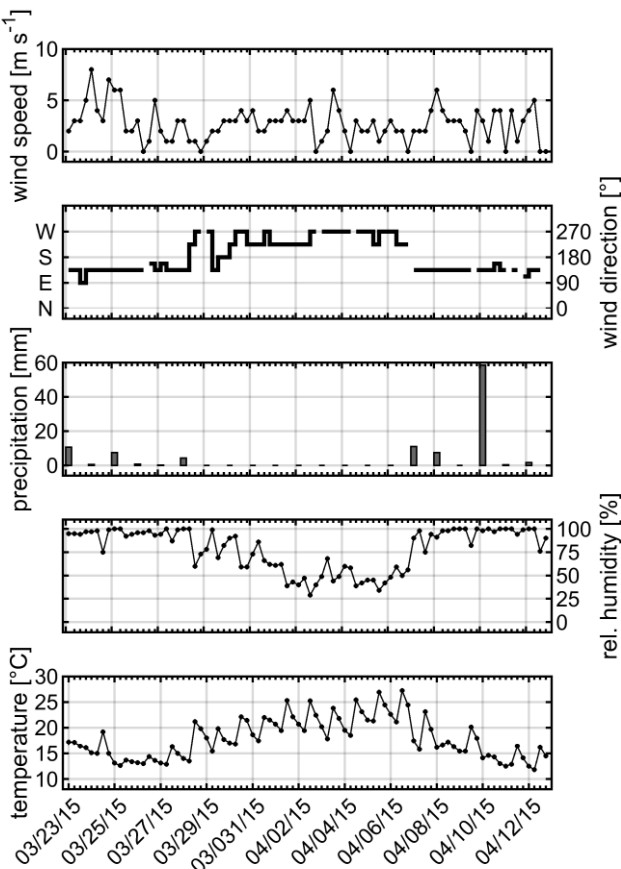

**Figure 2**. Time series (local time) of observed meteorological conditions at PDI during the sampling campaign from 23[rd] March to 12[th] April 2015.

**3.2 PM$_{2.5}$ bulk carbonaceous component and its organic compositions**

**3.2.1 Organic carbon (OC), elemental carbon (EC) and levoglucosan (LEV)**

Figure 3a displays the temporal variations of organic carbon (OC), elemental carbon (EC) and levoglucosan concentrations determined in the daily aerosol particle samples. OC and EC concentrations span ranges of more than one order of magnitude from 1.8 to 38.3 µg m$^{-3}$ (OC) and 0.13 to 17.8 µg m$^{-3}$ (EC), respectively, which are comparable with those measured in the North Southeast Asia region during dry season, i.e. Sonla, Vietnam in 2012 and 2013 (OC: 1.0 – 40.6 µg m$^{-3}$, EC: 0.0 -7.1 µg m$^{-3}$, Lee et al. (2016); Phimai, central Thailand in 2006 (OC: 2.3 – 16.7 µg m$^{-3}$, EC: 0.0 - 6.6 µg m$^{-3}$, Li et al. (2013)); Suthep Mountain, Chiang Mai, Thailand in 2010 (OC: 5.1 – 26.8 µg m$^{-3}$, EC: 1.6 -10.4 µg m$^{-3}$, Chuang et al. (2013)); Doi An Khang, Thailand in 2015 (OC: 20.0 – 75.6 µg m$^{-3}$, EC: 3.1 -11.1 µg m$^{-3}$, Pani et al. (2019a)). Highest concentrations of both OC and EC were observed on 5[th] and 6[th] April. The temporal trend of OC and EC correlated with the observed levoglucosan (LEV), which is known as a marker of BB (Simoneit et al., 1999), pointing towards substantial influence of BB on the carbonaceous aerosol content. The data are also summarized in Table S1 as average concentrations of OC, EC, and together with further constituted organics and classes in Figure S1 and Figure S2.

The ratios of OC to EC have been widely used to derive information about emission sources (Chow et al., 2004b; Han et al., 2010). Ratios of about 5 were reported in the winter in Xi-an, China, which was attributed to coal combustion for residential heating and BB; on the other hand, low ratios about 1 were attributed to vehicle and traffic-related sources. BB –influenced aerosol OC-to-EC ratios were 4.8 in Phimai, Thailand (Li et al., 2013) and ~ 6 in Chiangmai, Thailand (Chuang et al., 2013; Pani et al., 2019a), and ~ 6 in Sonla, Vietnam (Lee et al., 2016). In our study, the ratio averaged at 4.8, which is comparable with BB-influenced aerosol in Southeast Asia. However, we emphasize that OC/EC larger than unity only indicates the absence of efficient combustion as dominating OC and EC source and does not permit a direct assignment to BB. We take two approaches to learn about the origin of the carbonaceous aerosol based on OC, EC and LEV. Firstly, we link LEV to OC, and OC to EC, in order to derive primary OC$_{BB}$ and secondary OCsec. Secondly, we use char- and soot-EC to examine the sources of EC.

3.2.1.1 Simple OC source apportionment

Considering the ratios of LEV to OC, an estimation of the contribution of primary BB-derived OC (OC$_{BB}$) to ambient OC can be made according to

$$OC_{BB} = (LEV/OC)_{ambient} / (LEV/OC)_{source} \qquad (2)$$

LEV/OC may considerably vary between different BB aerosol sources, hence an average value of 8.14 % has been used for (LEV/OC)$_{source}$ (Wan et al., 2017). The results span an interquartile range from 35.7 % to 53.0% with a median of 44.2 % and peaking contribution on 3[rd] April (Figure 3b), showing that primary OC$_{BB}$ was the major source of OC during the sampling campaign and higher than 8.85 - 35.2 % during the pre-monsoon time in the Indo-Gangetic Plain (Wan et al., 2017). A second approach using OC data enables estimating the amount of secondary OC (OC$_{sec}$) based on excess OC from the minimum ratio of OC to EC (OC/EC)$_{min}$ (Turpin and Huntzicker, 1995), which was 2.15 and coincided with the highest OC concentrations on 6[th] April.

$$OC_{sec} = OC - EC \cdot (OC/EC)_{min} \qquad (3)$$

An interquartile range from 13.9 % to 50.0 % with a median of 22.1 % (Figure 3b) was obtained, suggesting that atmospheric aging is also an important source of OC at PDI. However, on three days the sum of the relative contributions of OC$_{sec}$ and OC$_{BB}$ add up to more than 100%. This might be a result of an underestimated (LEV/OC)$_{source}$ and/or overestimated (OC/EC)$_{min}$.

Moreover, the ratio of LEV to total carbon (TC) on 6[th] April accounted for 0.03. Within the range of LEV/TC for various BB emissions compiled by Zhang et al. (2015), this result appears at the lower limit softwood combustion and close to burning of agricultural residues, such as rice straw being typical for this region.

### 3.2.1.2 Simple EC source apportionment

In our further analysis, we operationally defined char-EC as $EC1_{apparent}$ - OP and soot-EC as EC2+EC3 according to Han et al. (2007) in order to examine the source of EC. At low concentrations of EC, soot-EC account for 50% of EC species, whereas at high EC concentrations, the fraction of char-EC increases and becomes dominant (Figure 3c). Soot-EC is considered to be more associated with motor vehicle emissions and char-EC with the combustion of solid fuels, such as biomass or coal (Han et al., 2010), indicating changing emission sources of EC and an association of high EC concentrations with solid fuel combustion during the sampling period. Furthermore, three distinct groups may be recognized from the EC species: (1) 12 days with less than 2.0 µg m$^{-3}$ char-EC; (2) 6 days with char-EC between 2.0 µg m$^{-3}$ and 7.0 µg m$^{-3}$; and (3) two days with more than 7.0 µg m$^{-3}$ char-EC. Since char-EC and levoglucosan are highly correlated (Pearson's r of 0.96), we derive that during the sampling campaign there were three periods with varying degrees of BB influence on PM$_{2.5}$ composition.

**Simple Source Apportionment from OC and EC**

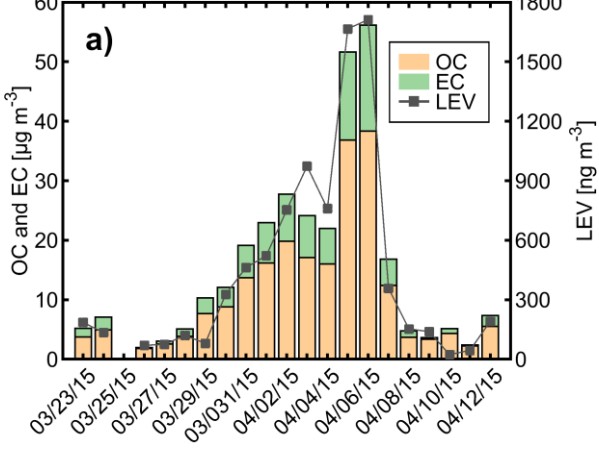

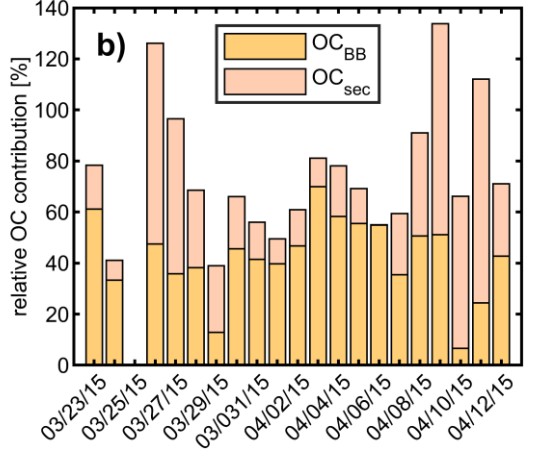

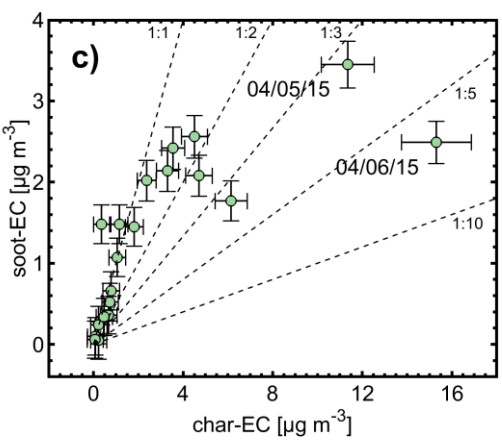

230

**Figure 3. a)** Temporal variations of OC, EC and levoglucosan (LEV) as BB marker in $PM_{2.5}$ at PDI during the sampling campaign from 23[rd] March to 12[th] April 2015; **b)** Relative contributions of primary OC from BB ($OC_{BB}$) and secondary organic carbon formation ($OC_{sec}$); **c)** Scatterplot of char-EC vs soot-EC during sampling campaign. Error bars represent measurement uncertainty calculated with formula (1) with precision and LOD of 10% and 0.36 µg $m^{-3}$ for char-EC and 5% and 0.23 µg $m^{-3}$ for soot-EC, respectively. Dashed lines indicate the ratio of char-EC to soot-EC.

235

### 3.2.2 Cluster analysis of Organic Aerosol Composition

We grouped the data from chemical OA speciation into the classes PAH, o-PAH, anhydrous sugars (AS), methoxyphenols, nitrophenols, fatty acids, and n-alkanes and performed together with OC a hierarchical cluster analysis in order to examine if the broadly defined OA composition follows the same trend as OC, EC and levoglucosan. A clustergram (Figure **4**) was used to illustrate days and classes related to organic particle constituents with similar composition and temporal behavior, respectively. A clustergram consists of a heatmap in the center with two dendrograms in horizontal and vertical position, generally illustrating similarities of variables and observations. The dendrogram of the upper part of the clustergram suggests three major time periods with distinctly different OA composition. Based on the previous section, we label the days "low BB-influence", "medium BB-influence", and "high BB-influence". The dendrogram on the left hand side emphasizes which variables are best representing the classification of the days and give insights why individual days stick out within one of the three major clusters. For both dendrograms in the clustergram, the cophenetic correlation coefficient is >0.95, showing excellent representation of the pairwise distances within variables and within observation. It can be derived that the BB-related compound classes of methoxyphenols and anhydrosugars seem to originate from the same source and to a lesser extent fatty acids and alkanes. Since OC, PAHs and o-PAH do not follow the same trend as the methoxyphenols and anhydrous sugars, we can expect significant contribution from other sources, such as traffic or in the case of elevated levels of o-PAH and OC, atmospheric aging. In contrast, nitrophenols show a different temporal behavior, thus appearing on an isolated branch of the dendrogram. Considering the attribution of primary and both primary and secondary origin to all other compound classes, nitrophenols may be predominantly formed by atmospheric aging. Of all considered compound classes, nitrophenols on 5[th] April had the largest absolute distance of 4.1σ from the campaign mean, which may be a consequence of high concentrations of the primary BB-derived monoaromatic compounds and their transformation into nitrophenols during few days and nights of aging (e.g. as described in Wang et al. (2020). We keep the classification of the sampling days by the clustergram for further discussion of the data in the following sections.

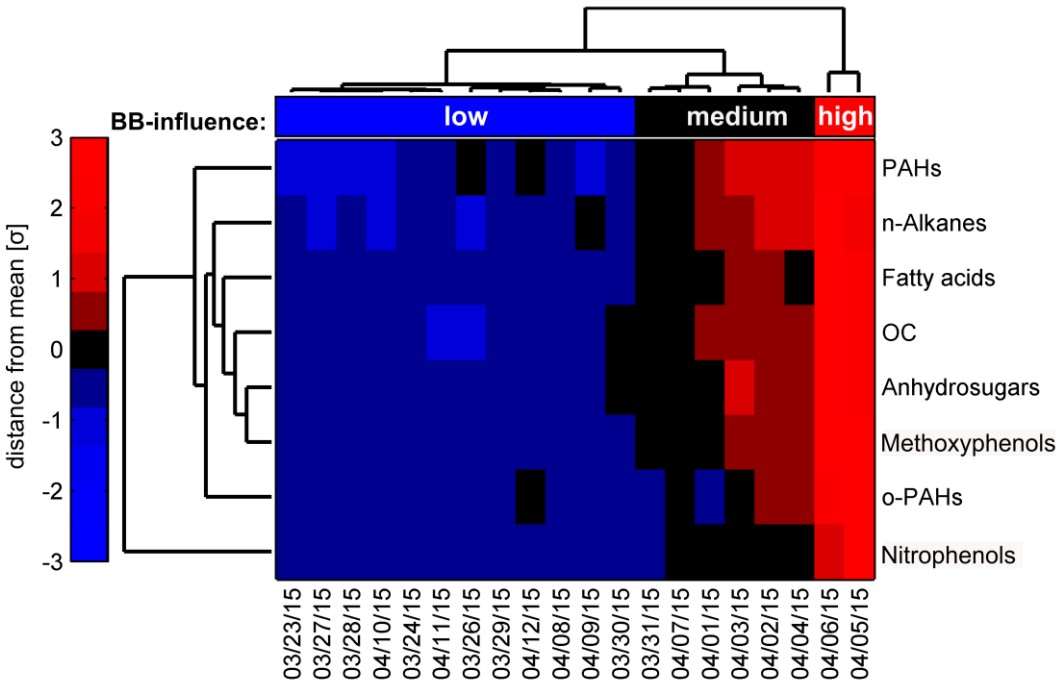

**Figure 4**. Clustergram based on organic composition of PM$_{2.5}$ sampled at PDI during the sampling campaign from 23$^{rd}$ March to 12$^{th}$ April 2015. Dendrograms indicating organic compounds and OC can be seen at the top and sampling days clustering on the left of the diagram. The colorbar on the left indicates the shades associated with values in the cells and denotes the distance to the mean in multiples of the standard deviation σ: Dark to bright blue denotes the lower values, dark to bright red the higher values and black refers to values closest to the mean.

### 3.2.3 Molecular composition of aerosol constituents

In this section, molecular constituents of OA and their possible origins are discussed compound class-wise and presented in Figure 5. Additional information is provided in the Supplement Figure S1-S9. In total, the speciated OA constituents account for up to 4% of OC.

#### 3.2.3.1 Anhydrosugars

Anhydrosugars, including levoglucosan (LEV), mannosan (MAN) and galactosan (GAL), produced by pyrolysis of cellulose and hemicellulose, are universal markers for BB (Simoneit and Elias, 2001). In the samples collected at PDI, LEV is the most abundant organic compound of the speciated OC ($23 - 1710$ ng m$^{-3}$, $0.22 - 2.5\%$ of OC). LEV abundance increases with increasing OC concentrations, indicating a significant impact of BB at receptor site. Ratios of LEV to MAN (LEV/MAN) may give an indication of the type of biomass that was burned. However, due to very high LEV amounts on the PDI filter samples, broad peaks in IDTD-GCMS analysis partially overlapped with MAN and GAL. This leads to high uncertainty of the quantitative results for the latter components. However, LEV and MAN quantities were also determined through additional HPLC-based measurements on parallel samples as detailed in Bukowiecki et al. (2019). The quantities are reported in Table S2. During the two days with high BB-influence, we obtain a ratio of LEV/MAN of 16.7 and 18.1 and also for medium BB-influenced days LEV/MAN ranges from 9.2 to 18.3, whereas days with low BB-influence have LEV/MAN between 1 and 7.5. Sang et al. (2013) compiled LEV/MAN for several types of biomass burning with evidence for low LEV/MAN ($4.0\pm1.0$) for softwood, medium LEV/MAN ($21.5\pm8.3$) for hardwood and high LEV/MAN ($32.6\pm19.1$) for crop residue burning; the latter category may be expanded to plants with a low level of lignification, including gramineae. Hence, it appears that softwood burning is the dominant BB source during days of low BB-influence, whereas PM2.5 collected during days of medium and high-BB influence must have contributions from the combustion of hardwood and/or plants with low level of lignification. Additional determination of the stable isotopes in AS might allow a more specific identification of the BB source, such as woody biomass or crop residues (Sang et al., 2012).

#### 3.2.3.2 Methoxyphenols

Methoxyphenols are products from the pyrolysis of biomass containing lignin, a constituent of all vascular plants present mostly between cellular structures and cell walls, and admit to differentiate between angiosperms, gymnosperms and gramineae (Simoneit, 2002). Vanillin and vanillic acid are markers for conifers (gymnosperm), whereas syringaldehyde, syringic acid are released from the combustion of angiosperms (and also found minor amounts in gymnosperm smoke.), and m-/p-hydroxybenzoic acid and acetosyringone are characteristic for gramineae (referring to grasses and non-woody vegetation).

During the sampling period at PDI, m-, p-hydroxybenzoic acid and acetosyringone appeared in the range of 0.8 - 247 ng m$^{-3}$ and 0.1 - 49.0 ng m$^{-3}$, respectively, which may be linked to agricultural residue burning practice in the northern Southeast Asia. Syringaldehyde, and syringic acid were in the range of at 0.0 - 40.2 and 0.1 - 47.9 ng m$^{-3}$, respectively, whereas vanillin and vanillic acid were found in lower quantities ($0.2 - 11.8$ ng m$^{-3}$, and $0.1 - 49.1$ ng m$^{-3}$). During the days of low-BB influence, gymnosperm-related methoxyphenols of the vanillyl-type showed higher relative abundancies than during the other two periods, which agrees well with low observed values for LEV/MAN. Vice versa, days of medium and high BB-influence had a larger share of syringyl-type methoxyphenols and hydroxybenzoic acids, but with ambiguous temporal trends. The ratio of syringic acid (SYAH) to vanillic acid (VAH) has been proposed as a diagnostic ratio to distinguish between different types of BB with 0.01 to 0.2 for gymnosperm burning and 0.1 to 2.44 for woody and non-woody angiosperm burning (Myers-Pigg et al., 2016). Across the entire sampling period, we observed $0.3 < SYAH/VAH < 1.7$ (Figure S3), giving evidence for dominating angiosperm burning. The peak of SYAH/VAH appeared on 4[th] April, which still belongs to the period of medium

BB-influence, right before transition to high BB-influence. Taking LEV/MAN and high concentrations of hydroxybenzoic acids into consideration, hard wood burning and in particular burning of non-woody grasses such as in agricultural residue burning may be the main types of BB observed at PDI.

Atmospheric aging alters the composition of primary BB plumes. It may therefore also affect diagnostic ratios. However, also a change in the contributions of different types of BB may have a similar effect. Myers-Pigg et al. (2016) reported the ratio of syringic acid (SYAH) to syringaldehyde (SYA) as a useful metric to estimate the freshness, i.e. degree of atmospheric aging, of a BB plume. During days with medium and high BB-influence, SYAH/SYA did not drop below 0.81 and revealed highest values of 2.75 and 3.44 on 5[th] and 6[th] April, respectively, indicating rather fresh plumes and low influence of atmospheric aging (Figure S4).

### 3.2.3.3 High Molecular Weight (HMW) n-Alkanes (C20:C33)

The daily summed concentrations of targeted n-alkanes ranged from 7.2 to 262 ng m$^{-3}$ during the sampling period at PDI. The concentrations were in the range from 7.2 to 45 ng m$^{-3}$ within days of low BB-influence and concentrations up to 262 ng m$^{-3}$ for days with high BB influence. We calculated the Carbon Preference Index (CPI) from the relation of n-alkanes with odd and even carbon number and determined the most abundant n-alkane for each day ($C_{max}$). Originally, the CPI for higher plant wax was calculated using n-alkanes from C21 to C34 (Simoneit, 1989). However, because of missing data for C34, we modified the CPI for this study by including n-alkanes from C21 to C32 to CPI$_{21-32}$.

Generally speaking, the main source of n-alkanes is epicuticular plant growth showing a strong dominance of odd-numbered carbon chain lengths with a maximum concentration ($C_{max}$) at $C_{29}$ and $C_{31}$. Another possible source of n-alkanes are vehicle emissions with a maximum abundance at $C_{25}$ for gasoline and $C_{20}$ for heavy duty within the range of $C_{19}$ to $C_{32}$ (Simoneit, 1986). Emissions from the combustion of fossil fuels have typically a CPI of about 1, while emissions of biogenic origin (e.g. plant abrasion, emissions from biomass combustion) often have a CPI of >2 (Simoneit and Mazurek, 1982).

CPI values for days with low BB influence ranged from 0.95 to 1.6, which indicated a significant contribution of fossil carbon sources (Cohen et al., 2010) to the organic matter despite the remote location of the sampling station, but with lowest concentrations of n-alkanes measured during the sampling period at PDI. On the other hand, CPI values for days with medium- and high BB-influence were from 1.5 to 2.8, inferring together with $C_{max}$ of 29 the input from biogenic sources and suspension of debris from BB (Fig. S4).

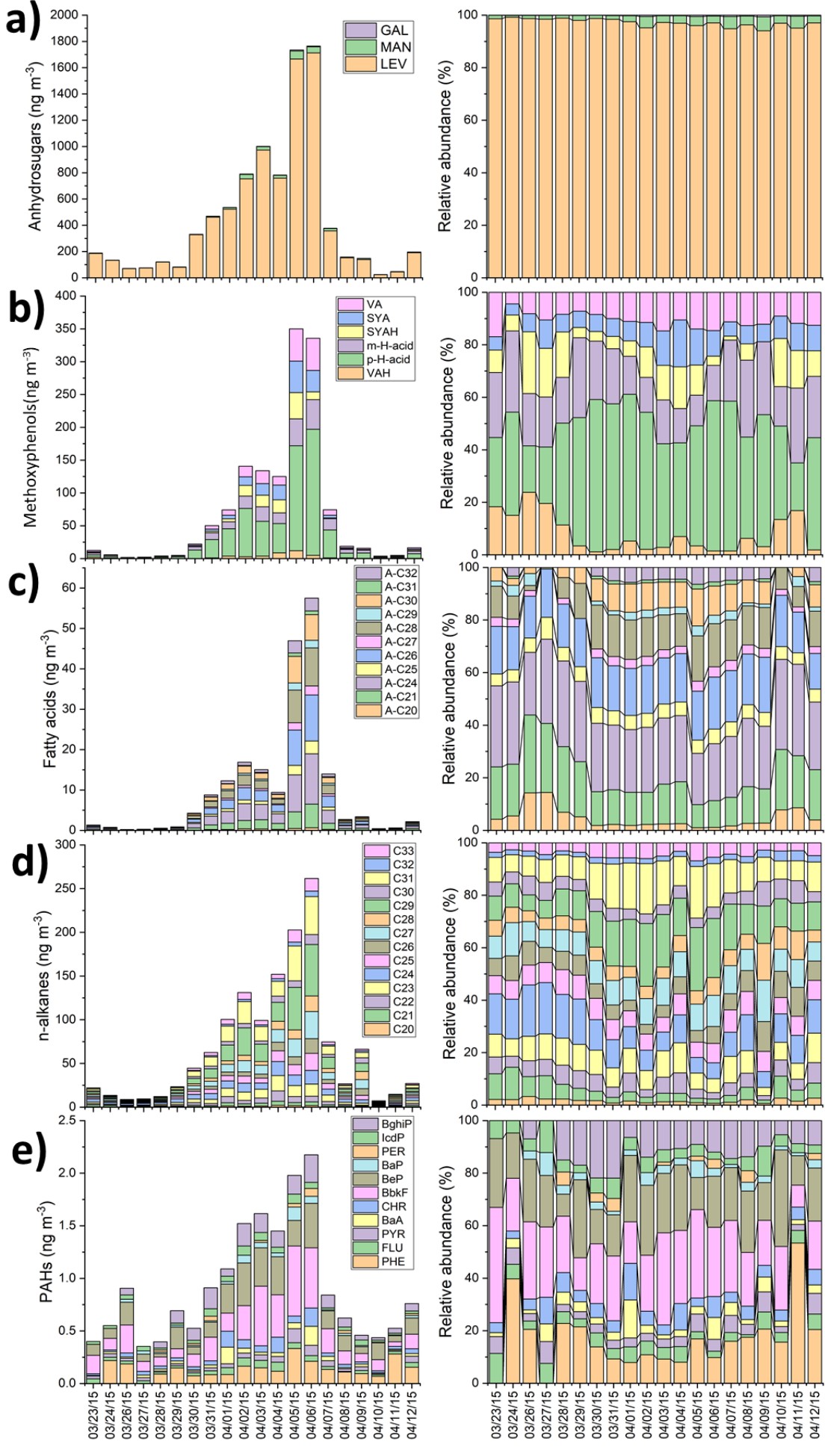

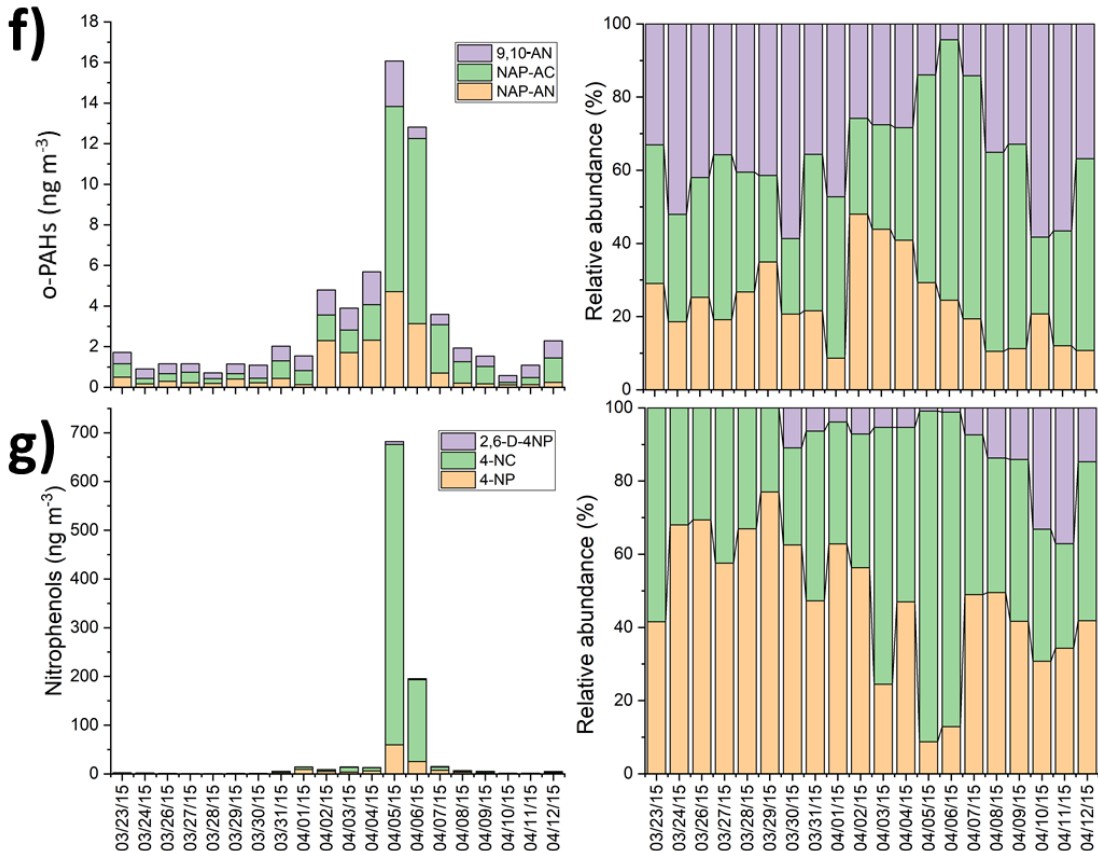

**Figure 5.** Time series of organic constituents in daily aerosol particle samples (n=21) at PDI during the sampling campaign from 23rd March to 12th April 2015. a) anhydrosugars, b) methoxyphenols, c) HMW fatty acids, d) n-alkanes, e) PAHs, f) o-PAHs, g) nitrophenols. Left column provides mass concentrations (ng m$^{-3}$) and right column provides relative mass fractions within the compound class.

### 3.2.3.4 High Molecular Weight (HMW) Fatty Acids (C20:C32)

HMW fatty acids are known to be released released from terrestrial higher plant waxes, but also from different kinds of BB, cooking and vehicular emissions (Ren et al., 2020). In general, during BB events, lipid-related compounds that are originally present as leaf-waxes can be emitted to the air together with smoke particles via volatilization and recondensation onto particulate matter without oxidative degradation. Enhancement of HMW fatty acids concentration are reported to coincided with levoglucosan and HMW n-alkanes during intensive wheat straw burning periods in Mt. Tai, northern China and Nanjing, East China (Wang et al., 2009;Fu et al., 2012). The daily summed concentrations of HMW fatty acids at PDI ranged from 0.2 – 57.5 ng m$^{-3}$ during sampling period, in which the low range (0.2 – 3.4 ng m$^{-3}$) of concentration took place at the beginning and at the end of sampling period, belonging to low-BB days. HMW fatty acids with even carbon number showed higher concentrations than the ones with odd carbon number and a fairly stable ratio of 5 to 7, pointing toward domination of biogenic or BB origin. Most of the individual HMW fatty acids at PDI were below the limit of quantification for more than 50% of the days, so temporal trends are only examined for their total concentration. A linear trend could be observed for total HMW fatty acids vs. total AS concentrations for days with low and medium BB-influence. However, the extrapolation of the linear fit from Deming regression to days of high BB-influence would underestimate the observed HMW fatty acids concentrations (Figure S5), suggesting a different combustion source or additional source such as possibly the atmospheric oxidation of n-alkanes or n-alkenes.

### 3.2.3.5 PAHs

Eleven PAHs, which are known as markers for incomplete combustion, were quantified from the samples collected at PDI: phenanthrene (PHE), fluoranthene (FLU), pyrene (PYR), benz[a]anthracene (BaA), chrysene (CHR), sum of benzo[b]fluoranthene and benzo[k]fluoranthene (BbkF), benz[e]pyrene (BeP), benz[a]pyrene (BaP), perylene (PER), indeno[1,2,3-cd]pyrene (IcdP), and benzo[ghi]perylene (BghiP). The sum concentrations of detected PAHs cover the range of $0.35 - 2.17$ ng m$^{-3}$ and $0.26 - 1.95$ ng m$^{-3}$ without considering PHE, respectively, which is sensitive to sampling conditions because of its volatility. Among the targeted PAHs, PER showed the lowest average concentration (range: $0.00 - 0.07$ ng m$^{-3}$), while the BbkF showed the highest average concentration (range: $0.04 - 0.67$ ng m$^{-3}$). All individual PAHs reached their highest concentrations during the 05[th] and 06[th] April with high suspected influence of BB (Figure 5).

Comparing total PAH concentrations in the vicinity of the study area, the resolved PAHs were much lower concentrated than those found in the BacGiang and HaNam, rural sites in north of Vietnam ($80 - 2200$, and $170 - 800$ ng m$^{-3}$ respectively) (Anh et al., 2019), but comparable with those found in Chiangmai, Thailand in dry season ($0.3 - 6.8$ ng m$^{-3}$) (Chuesaard et al., 2013). The ratio of some selected PAHs can be useful to roughly discriminate PAH sources. During the entire campaign, there was a noticeable influence of pyrogenic (combustion-derived) PAH to ambient PM2.5, indicated by the median diagnostic ratio of 0.62. However, neither the diagnostic ratios for combustion sources follow the trends of molecular markers for BB, such as levoglucosan or lignin monomers, nor do they show elevated values for the days of high BB influence (Figure S6). In the diagnostic ratio BaP/(BaP+BeP), the faster degradation of BaP in the atmosphere is exploited with values below 0.5 indicating degradation of the aerosol by photochemical aging (Tobiszewski and Namiesnik, 2012). However, the BaP/(BaP+BeP) is unknown for the fresh emission and recent studies revealed that BaP/(BaP+BeP) for residential heating (Miersch et al., 2019; Vicente et al., 2016) may fall below the threshold of 0.5. Apart from three days, all BaP/(BaP+BeP) were below 0.15, pointing towards substantial atmospheric aging. The maximum of 0.35 was observed on 5[th] April, which belongs to one of the two days of high BB-influence (Figure S6). This indicates a fairly nearby combustion source of PAH. However, other specific values of PAH diagnostic ratios for IcdP/(IcdP+BghiP), BaA/(BaA+CHR) and FLU/(FLU+PYR) during the sampling period at PDI appear outside the range suggested for vegetation fires or wood burning (Galarneau, 2008; Tobiszewski & Namiesnik, 2012) on any day. For example, during the entire campaign at PDI the diagnostic ratio IcdP/(IcdP+BghiP) constantly appears below the value of 0.5 although values exceeding 0.5 are suggested as an indicator of grass, coal and wood combustion. Consequently, it did not lead to further evidence for BB as dominating aerosol source, which agrees with the relatively large distance of total PAH to BB markers such as anhydrous sugars and methoxyphenols in the clustergram (Fig. 4). However, there is a clear trend between total 4-ring to 7-ring PAH and AS with a Pearson correlation coefficient of 0.941. Moreover, slopes and intercepts obtained from Deming regression for either only days of low and medium BB-influence or over the entire sampling period at PDI are not significantly different at a significance level of 5%, indicating that PAH concentrations are predominantly associated with BB (Figure S7).

This disagreement between source apportionment by PAH diagnostic ratios and other molecular markers used in this study may have several reasons. First, there is a considerable variation of PAH in combustion emissions which might be used for a limited number of PAH sources, combustion conditions or fuel properties. Secondly, as exploited in BaP/(BaP+BeP), individual PAH of proposed diagnostic ratios may have different rate constants towards atmospheric oxidants, so even major primary emission source can be less ambiguously identified. Furthermore, major differences in PAH patterns at PDI arise from semi-volatile three- and four-ring PAH PHE, FLU and PYR, which may be caused by dynamic gas-particle partitioning. Katsoyiannis et al. (2011) demonstrated that using PAH diagnostic ratios from emissions at a site close to a highway were classified to originate from non-traffic sources. The classification of the days during the sampling period at PDI rather corresponds to the total concentrations of PAH, which were found to be significantly different at a significance level of 5% between low-, medium- and high BB periods using a one-way analysis of variance (ANOVA) with Bonferroni post-hoc correction. Altogether, PAH diagnostic ratios seem to be less specific than other

molecular markers discussed for BB and do not increase the knowledge about the sources and processes relevant for the PM2.5 composition at PDI during the sampling period.

### 3.2.3.6 Oxygenated PAH (o-PAHs)

Three o-PAHs have been quantified in the samples from PDI: 9,10-Anthracenedione (9,10-AN), 1,8-naphthalic anhydride (NAP-AN), naphthoic acid (NAP-AC). o-PAHs are associated with incomplete gasoline, diesel, coal or wood combustion as well as with the secondary formation from PAH precursors (Walgraeve et al., 2010). The total concentrations of the three o-PAHs varied between 0.19 ng m$^{-3}$ and 5.36 ng m$^{-3}$, peaking during the days of high BB-influence on 5$^{th}$ and 6$^{th}$ April. Among the o-PAHs, 9,10-AN showed the lowest average concentration (0.75 ng m$^{-3}$), while NAP-AC showed the highest average concentration (1.62 ng m$^{-3}$). It seems that 9,10-AN is less associated with the observed BB plume due to its relatively low concentration range and distinct lower relative contribution to total o-PAH concentrations during days of high BB-influence. At the beginning of the medium-BB period, NAP-AN had its highest relative abundance during the sampling campaign and declined on the following days while NAP-AC steadily increased, which moderately correlates with the ambient temperature (Figure 2). In the aqueous phase, NAP-AN may be rapidly hydrolyzed to NAP-AC depending on the temperature and pH (Barros et al., 2001). Therefore, we used the sum of NAP-AN and NAP-AC for source apportionment at PDI.

For the days of low and medium BB-influence, we found a Pearson correlation coefficient of 0.8 between the sum of NAP-AN and NAP-AC and AS, indicating predominantly primary origin. However, the expected concentrations of NAP-AN and NAP-AC from a linear fit by Deming regression accounts for only 50% of the observed concentrations (Figure S8). Hence, NAP-AN and NAP-AC on days with high-BB-influence either originates from a combustion source with different biomass and/or combustion conditions, or to similar extent from a common BB and another source such as secondary formation. In contrast to NAP-AN/-AC, 9,10-AN showed a linear correlation to AS over the entire sampling period with the exception of 6$^{th}$ April and is thus predominantly of origin from primary BB. Considering the peaking OC on 6$^{th}$ April and often higher reactivity of o-PAHs compared to their structural PAH analogues (Ringuet et al., 2012), lower concentrations of 9,10-ANT in the concentration range of days with low-BB-influence might be a consequence of a lower influence of atmospheric aging compared to plumes arriving at PDI on 5$^{th}$ April..

### 3.2.3.7 Nitrophenols

Nitrophenols may generally originate from various primary sources, e.g. traffic, coal and biomass combustion, herbicide and pesticide usage, but are also formed by reactions of monoaromatic compounds with nitrate radicals (NO$_3$) or hydroxyl (OH) radicals in the presence of NO$_x$ (Harrison et al., 2005; Li et al., 2016; Kahnt et al., 2013). In particular, nitrocatechols refer to an important subclass of nitrophenols which belong to the group of UV-light-absorbing species and are known to affect the radiative balance and climate of Earth (Laskin et al., 2015).

In the samples from PDI, the compounds 4-nitrophenol (4-NP) and 4-nitrocatechol (4-NC) account for 0.4-60.0 ng m$^{-3}$ and 0.2-616.0 ng m$^{-3}$, respectively. Peak concentrations coincided with enhanced LEV abundance on 5$^{th}$ and 6$^{th}$ April, and reached enhancement factors up to 10 (4-NP) and 600 (4-NC) compared to other low and medium BB-influenced days during sampling campaign. The quantities of 4-NP in our study was lower than that of in haze episode in Shanghai (range 150-770 ng m$^{-3}$), but much higher than in a one-year study in Belgium (range 0.32 - 1.03 ng m$^{-3}$). In contrast, 4-NC in our study was higher than that of in Shanghai (22 – 154 ng m$^{-3}$) and in Belgium (0.49 - 9.0 ng m$^{-3}$) (Li et al., 2016; Kahnt et al., 2013).

Individual nitrophenols in ambient air have been recently attributed to their predominant source in a rural site in China, assigning NP to predominantly BB and NC to predominantly secondary formation (Salvador et al., 2021). We examined the relation of individual nitrophenols at PDI to AS, which are strongly associated with BB (Figure S9). For days of low and medium BB-influence, we obtained moderate correlations for all nitrophenols with AS. However, on days with high BB-influence the ratio of each individual nitrophenol to AS was higher than on days with low and medium BB-influence many times over. Hence, nitrophenols have a different source on days of high-BB influence, which might be a combustion source

with different biomass or combustion conditions or atmospheric aging. However, higher concentrations of 4-NC point towards formation by atmospheric ageing as NC was found to originate to a greater extent from this source than BB (Salvador et al., 2021). Regarding 4-NP, it has been found that concentrations of 4-NP in wood combustion emissions aged in an oxidation flow reactor have a maximum after short aging of approximately 1-2 equivalent days because reactivity towards OH radicals and photolysis (Hartikainen et al., 2020). Despite similar concentrations of AS on both days with high BB-influence, the concentration of 4-NP, 4-NC and 2,6-D-4-NP on 5[th] April were 2.4-fold, 3.7-fold and 2.7-fold higher than on the 6[th] April. Additionally, the BB plume on 5[th] April arrived at PDI in early morning, as indicated by lowest MCE values (available on 1 hour time resolution) during the campaign (see section 3.4. and Figure S10). It has been reported that the formation of 4-NC from catechol proceeds approximately three times faster during night than during daytime (Finewax et al., 2018). Therefore, a possible explanation of elevated nitrophenol concentrations in the high-BB episode on 5[th] April might be associated with BB plumes that aged overnight before arriving at PDI. On 6[th] April, however, atmospheric aging might have been of less importance for nitrophenol concentrations at PDI, which agrees with the lowest ratio of OC/EC and associated estimated contribution of $OC_{sec}$ to ambient OC (section 3.2.1.).

### 3.3. Trace gases observations and implications

**Fehler! Verweisquelle konnte nicht gefunden werden.** shows the time series of hourly averaged CO, $O_3$, $CO_2$ and $CH_4$ mixing ratios during the sampling campaign at PDI. The temporal evolution of these records also indicates the different conditions that were observed during the campaign.

CO mixing ratios were in the range of 220 – 600 ppb at the beginning and the end of the sampling campaign, but increased slowly from 31[st] March before reaching up to 1270 ppb on 5[th] April. As shown by Bukowiecki et al. (2019), since the beginning of the monitoring in 2014, each year from February to May CO levels at PDI are systematically enhanced and do not represent background mixing ratios as expected to be observed at a pristine sampling site. The latter is the case from June to September when CO mixing ratios below 100 ppb are observed The enhancement in spring can be explained by the large-scale BB in Southeast Asia (Lin et al., 2013) and particularly all over the Indochina peninsula in spring (Yen et al., 2013), which leads to high emissions of CO (Shi and Yamaguchi, 2014;Lelieveld et al., 2001). The observed variability and the additional increase by more than 800 ppb of CO during specific days of the 2015 intensive campaign at PDI confirms the hypothesis of advection of more regional BB events (Kondo et al., 2004;Pochanart et al., 2003).

After rainfall on 07[th] April, CO dropped back to the range observed before, but still did not reach typical values for the unpolluted boundary layer. Also $O_3$ shows a similar pattern like CO: mixing ratios had rather small variability in the beginning of the campaign before they increased to up to 90 ppb, followed by a sharp drop to only 4 ppb. $CO_2$ and $CH_4$ also show increases in their mixing ratios on 5[th] and 6[th] April, but with less variability. However, unlike CO and $O_3$, the mixing ratios of $CO_2$ and $CH_4$ remained elevated also after the days of high BB-influence and the following rainfall likely due to the north-easterly advection of other anthropogenic emission sources (see trajectory analysis of section 3.5.).

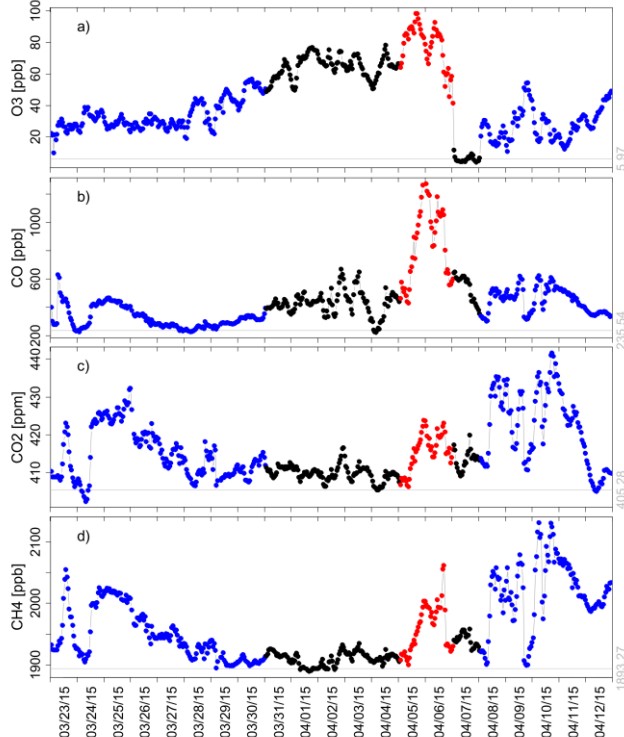

**Figure 6.** Time series of hourly a) ozone ($O_3$), b) carbon monoxide (CO), c) carbon dioxide ($CO_2$) and d) methane ($CH_4$) at PDI during the sampling campaign from 23[rd] March to 12[th] April 2015. Three colors, including green, black, and red follow the clustergram from statistical approach discussed in section 3.2.2. Grey lines in the plots indicate the background values, calculated as the mean of the lowest 5% of the dataset during the campaign.

Scatterplots of hourly averaged trace gases indicate different relationships between these gases during the sampling campaign (Figure 7. ). The correlation between CO vs $CH_4$ reveals two apparent branches, the upper (in red) was observed on 5[th] and 6[th] April when BB influence was high, the lower (in green) in the beginning and the end of the campaign. The distinction of the different BB influence regimes is best seen in the scatter plots with CO, which reflects the large contribution of BB to the

485 atmospheric CO burden, especially during the high BB influence days. Therefore, the gas trace observations confirm the classification of the sampling days based on the chemical composition of $PM_{2.5}$ in the clustergram.

Time series of $CO_2$, $CH_4$ and CO, being all predominantly emitted by primary sources, show different trends than $O_3$ which is mainly of secondary origin, highly reactive and also subject to deposition and removal. Therefore, the excess $\Delta O_3/\Delta CO$ ratio, utilizing a primary and secondary combustion gas, is a commonly used proxy to investigate the age of BB plumes (Jaffe and

490 Wigder, 2012) and to characterize $O_3$ production in smoke plumes. In general, $\Delta O_3/\Delta CO$ usually increases with increasing plume age of the BB plume due to net $O_3$ production (Parrington et al., 2013; Akagi et al., 2011; Jaffe and Wigder, 2012). $\Delta O_3$ and $\Delta CO$ were calculated by subtracting the "baseline" during the campaign which was determined as the average of the lowermost 5% of the data during the campaign (see the grey lines in Figure 7). However, reported values for the $\Delta O_3/\Delta CO$ ratio in aged plumes do not show a consistent pattern (between 0.1 and 0.9, Figure S10) for some plumes associated with

495 tropical fires (Andreae et al., 1994; Mauzerall et al., 1998) and boreal fires (Honrath et al., 2004; Bertschi and Jaffe, 2005) and as low as 0.1 in aged plumes from Southeast Asian studies (Kondo et al., 2004) as ozone formation also depends on the availability of nitrogen oxides and volatile organic compounds. To account for BB influence, we linked $\Delta O_3/\Delta CO$ ratios with levoglucosan during sampling campaign. During days with low BB influence, levoglucosan concentrations were in the range of 20-400 ng m[-3] and the $\Delta O_3/\Delta CO$ ratio increased from 0.05 to 0.8 (Figure S10), indicating other source contributions for $O_3$

production, i.e. photochemical conversion of biogenic VOC emissions (Nguyen et al., 2016). During days with medium BB influence, levoglucosan levels increased up to 1000 ng m[-3], and the excess ratio slightly decreased while being more variable. On 5[th] and 6[th] April with high BB influence, LEV concentrations reached up to 1.6 µg m[-3] and the excess ratio dropped below

0.1 especially on 6[th] of April, suggesting that the arriving BB plume was rather fresh. Right after the high BB plume event, $\Delta O_3/\Delta CO$ does not allow unambiguous interpretation because $O_3$ levelled off close to the background. The last part of the campaign, identified to be under low BB influence, is characterized by mostly low $\Delta O_3/\Delta CO$ ratios, which rose again toward the end of the campaign. It must be also emphasized that the calculation of the excess ratios is sensitive to the selection of the background levels, increasing uncertainty for $\Delta O_3/\Delta CO$ and associated plume age.

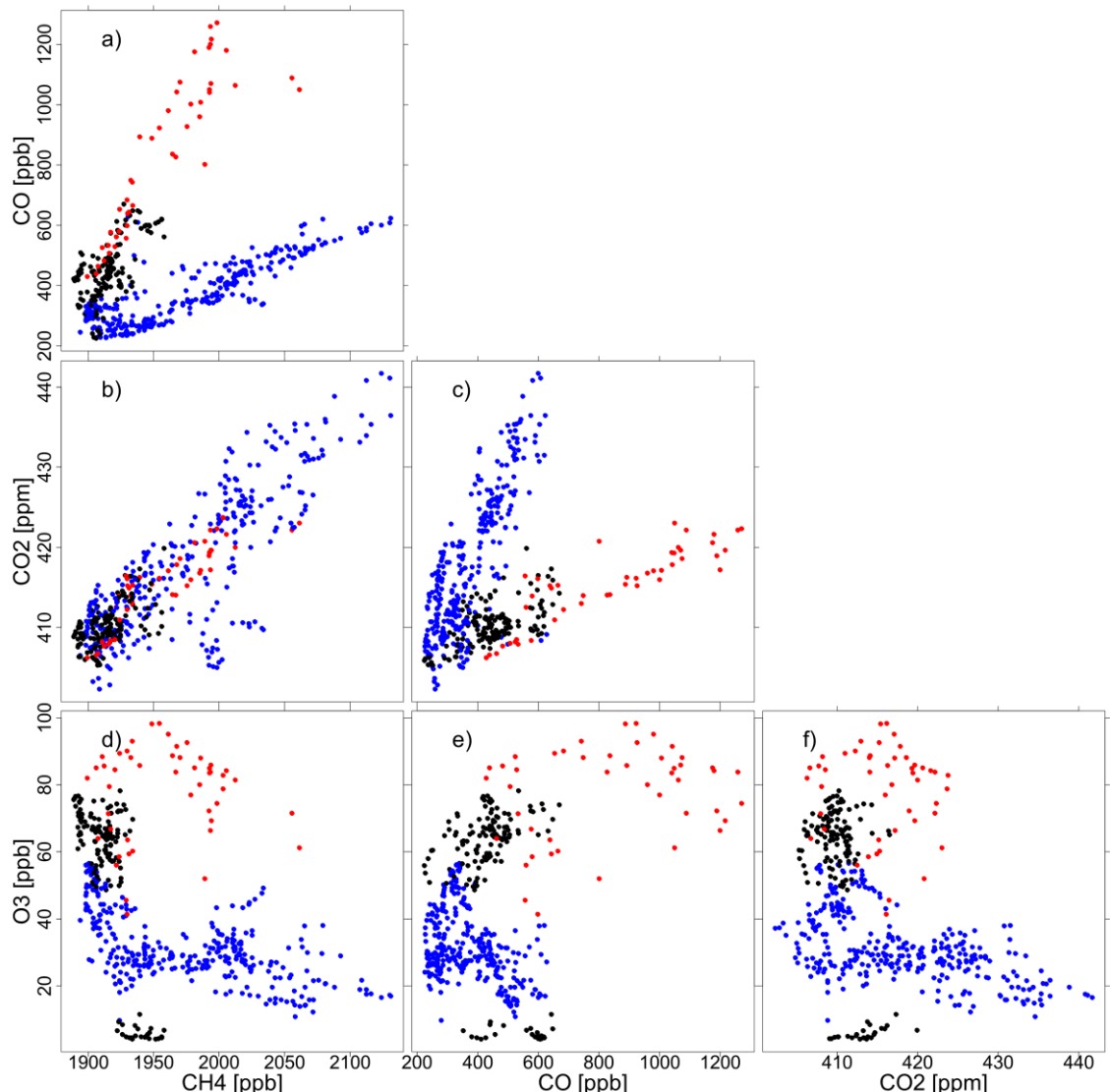

**Figure 7.** Scatter plots of hourly ozone ($O_3$), carbon monoxide (CO), carbon dioxide ($CO_2$) and methane ($CH_4$) vs. each other at PDI during the sampling campaign from 23[rd] March to 12[th] April 2015. Three colors, including blue, black, and red follow the clusters of low-, medium-, and high-influenced BB discussed in section 3.2.2.

### 3.4. Parametrization of Organic Aerosol (OA) constituents with MCE

The modified combustion efficiency (MCE), defined as

$$MCE = \Delta CO_2/(\Delta CO+\Delta CO_2) \qquad (4)$$

is widely used as an indicator of combustion emissions, with higher MCE values (close to 1) indicating higher proportions of flaming/complete combustion and lower MCEs referring to higher proportions of smoldering/incomplete combustion (Kondo et al., 2011). MCE values of 0.80 to 1.00 were observed for wildland fires in several vegetation zones (Akagi et al., 2011) and of 0.79 to 0.98 in regional smoke plume at the Mount Bachelor Observatory (Briggs et al., 2016). At PDI, except few hours on 23[rd] March, MCE was stable around 0.98 from the beginning of the sampling period and decreased to values below 0.94

from 30th March to 4th April. It occasionally reached values below 0.9 on 05th April, while it rises again back to 0.99 afterwards (see Figure S10). Variability in MCE at PDI was found particularly between 30th March and 6th April, which might be caused by variable degrees of mixing between the smoke plume and other air masses (Yokelson et al., 2013). The relatively high MCE values of the plume might be also a consequence of low precipitation over a longer period associated with biomass of low moisture, improving the combustion efficiency (Chen et al., 2010). For some days, both CO and $CO_2$ are close to the chosen background levels, so consequently, at $CO_2$ mixing ratios below 410 ppm, the uncertainty of MCE exceeds 0.04.

We examined the relation between MCE and the compound classes defined in Figure 4. Concentrations of all compound classes declined with increasing MCE due to more efficient combustion conditions (Figure 8) with a linear behavior as suggested by other studies considering a similar interval of MCE (Burling et al., 2010;Bertrand et al., 2017). To have null concentrations of OA at MCE = 1 (complete combustion), the slope of the regression should be equal to minus the intercept, which was in fact obtained from the regression (Table S3). PAHs, which are related to combustion processes, anhydrosugars and methoxyphenols, show the best agreement with the regression function and therefore a direct relation to the MCE. During the days of medium BB-influence, concentrations of other compound classes deviate from their regression function and are consistently underestimated (alkanes, o-PAHs, OC, fatty acids) because data points for days of high BB influence increase the steepness of the slope, indicating additional contributions by non-combustion sources such as vegetation and atmospheric aging. In particular the fit for nitrophenols is clearly dominated by the datapoints of the days of high BB influence. During the days with medium BB-influence, we observed on average 4.7-fold significantly higher concentrations ($p = 4 \cdot 10^{-6}$ from ANOVA) compared to low-BB-days, which is also associated with lower MCE. The MCE values for days of high BB-influence (0.941 and 0.952) appear within the interquartile range of the days with medium BB-influence (0.933 - 0.969), therefore giving evidence of atmospheric aging being the most important source of nitrophenols.

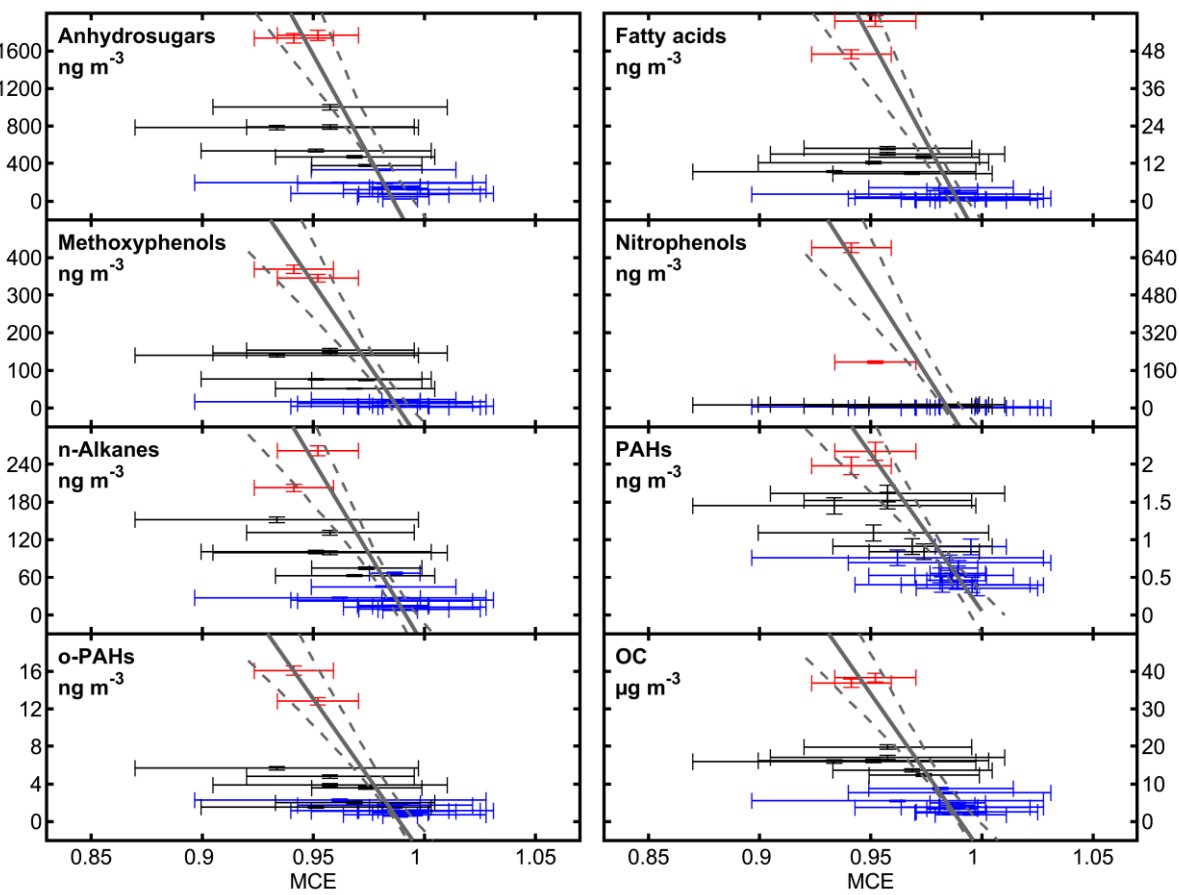

**Figure 8.** Relation between modified combustion efficiency (MCE) and various compound classes. Blue, black and red color correspond to the periods of low BB, medium and high BB-influence, respectively. Errorbars illustrate the uncertainty of each data point (please note that uncertainties for concentrations are partly too small to be visible in the figure). The gray lines refer

to linear fits from Deming regression. Dashed lines denote the non-simultaneous prediction band of the fit function by means of 1σ.

### 3.5. Back trajectory analysis

Average air mass backward trajectories for each of the periods identified by the organic aerosol clustering reveal a distinctly different advection pattern for the 3 pollution phases (Figure 8). Similar plots of individual trajectories (initialized on a 4-
550 hourly time step) can be found in the supplemental material (Figure S5). For the days with low BB-influence, atmospheric transport towards PDI was from north-eastern to eastern sector with a stronger tendency of trajectories arriving from mainland China after the polluted phase (April) as compared arriving over the Yellow sea before the polluted phase (March). Very little active BB was observed by MODIS in these regions. In clear contrast, air mass trajectories indicate the arrival of air from the south-western sector during the pollution event passing directly over large areas of active open BB in northern Laos (most
555 intense) and Myanmar (less intense) (Pani et al., 2019b;Lin et al., 2013). A main difference between the more polluted cluster (high) and the medium polluted cluster (medium) seems to be a recirculation over land area for the medium cluster as compared to a more westerly advection for the high cluster. On average air masses moved slightly faster in the high-BB cluster (larger distance between symbols in trajectory plot) than in the medium cluster. However, in both cases the area with the most intense fires was crossed within the last 24 to 36 hours before arriving at PDI, supporting the finding of little aging in the aerosol
composition as derived from $\Delta O_3/\Delta CO$. Nevertheless, $\Delta O3/\Delta CO$ ratios were lower in the high cluster as compared to the medium cluster, where variable ratios were observed. A possible explanation could be the longer residence time over land for the medium cluster, allowing to take up ozone precursors and to form ozone for a longer time, combined with an air mass origin over more $NO_x$ rich regions (southern China), in contrast to the slightly faster advection above the fires and an oceanic ($NO_x$ depleted) origin for the high cluster prior to arrival at PDI. Fire activities were more enhanced during the days of high
BB-influence than medium BB-influence (Figure S11), potentially explaining the observed differences in pollution loads. For all OA clusters, the average trajectories were travelling below 2 km above sea level in the last five days before arrival at PDI, suggesting that sampled air masses were in immediate contact with surface emissions.

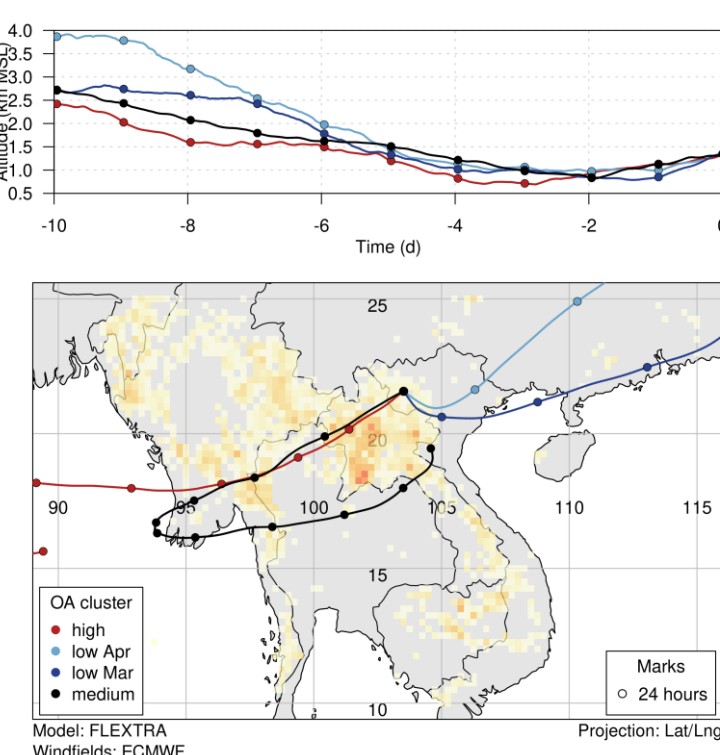

**Figure 8.** Ten-day backward trajectories arriving at PDI averaged for the periods determined by the organic aerosol clustering. The "low" class was additionally split into the period before the days with medium and high BB-influence (March: low Mar)

and after (April: low Apr). The upper panel displays the average height of the trajectories above sea level on a relative time axis before arrival at the site. The lower panel gives the average location of the trajectories overlaid on a map of MODIS fire count densities for the period spanning from five days before until the end of the investigation period (from low (bright yellow) to high (orange-red) fire intensities). Filled circles on the trajectories mark a travel time of 24 hours.

## 4. Conclusions

Our extensive characterization of carbonaceous $PM_{2.5}$ species and monitoring of trace gases at the Pha Din (PDI) station in Vietnam from 23$^{rd}$ March to 12$^{th}$ April 2015 gives insights into the atmospheric composition during the (dry) pre-monsoon season, which is impacted by large-scale open biomass burning (BB) on the Indochina Peninsula. OC and EC concentrations in $PM_{2.5}$ were found in the range from 1.8 to 38.3 µg m$^{-3}$ and 0.1 to 9.8 µg m$^{-3}$, respectively, which is comparable with other studies in the region during the dry season. Among several OC classes and 51 quantified organic aerosol (OA) constituents, anhydrous sugars and levoglucosan were the most abundant compound class and species, respectively. In a combined statistical and molecular marker approach, together with trace gases and backward trajectory (BWT) analysis, a consistent picture could be drawn that there were three distinct pollution periods with elevated levels of OA loadings associated with BB aerosol. The analyses of BWT, fire counts and $\Delta O_3/\Delta CO$ point toward a short plume age of less than 36 hours for the air masses of high BB-influence days and local/regional BB events by implication. However, daily averaged MCE > 0.90 remained high compared to literature values of other BB plume studies, which might be a consequence of the low precipitation amount during the dry pre-monsoon seasons, availability of biomass with low moisture content and finally improved combustion efficiency. In addition to the statistical classification of the sampling period, we applied several molecular marker and diagnostic ratio approaches to prove BB as major source of carbonaceous aerosol, to elucidate the type of BB and to estimate the role of atmospheric aging. Consistently, also the molecular OA composition points toward BB and suggests more precisely the combustion of angiosperm as main OC origin. While we did not find sufficient evidence for an unambiguously linkage to either wildfires or agricultural residue burning on days of high BB-influence, softwood burning appears of minor importance. Based on the ratio of levoglucosan to total carbon, agricultural residue burning is more likely than hardwood burning. Atmospheric aging was found to have a higher contribution to OC during days with low-BB influence than medium or high BB-influence. However, some OA constituents, such as nitrophenols and 9,10-anthracendione, which are known early-generation products of atmospheric aging of primary BB-related precursors, differed greatly from the expected concentrations based on their ratios to anhydrosugars. Therefore, we suggest that the plume arriving at PDI on 5$^{th}$ and 6$^{th}$ April underwent only a low degree of atmospheric aging and might have had a more local source.

Overall, our results agree with a previous source apportionment study based on optical aerosol properties by (Bukowiecki et al., 2019), and add valuable data on the OC chemical composition of $PM_{2.5}$ in a region of scarce data availability. These data provide valuable insights for a better characterization of OA emitted from BB in Southeast Asia and may be considered as a reference dataset also for future studies over more extended periods, or including more chemical parameters in other regions. The presented gas- and particle-phase data may also be used in the evaluation of atmospheric transport simulation models, or for comparison with BB emissions in other world regions and from other BB types, such as for instance Australian Bush Fires, African Savannah Fires, or Tropical Peatland Fires.

### Acknowledgement

This work was financially supported via research grant No. 91614707 by German Academic Exchange Service (DAAD) and No. QTRU05.01/18-20 by Vietnam Academy of Science and Technology (VAST). HC thanks for funding by the Helmholtz International Lab *aeroHEALTH* (InterLabs-0005) through the Federal German Helmholtz Association of Research Centers (HGF) and the Helmholtz Zentrum München. The continuous aerosol optical property and trace gas observations were setup and operated with support of the Federal Office of Meteorology and Climatology MeteoSwiss through the project Capacity

Building and Twinning for Climate Observing Systems (CATCOS) Phase 1 and Phase 2, Contract No. 81025332 between the Swiss Agency for Development and Cooperation (SDC) and MeteoSwiss. We thank the Vietnam Meteorological and
Hydrological Administration (VNMHA) for providing access to the facilities and for the support of our measurements. MS acknowledges funding from the GAW Quality Assurance/Science Activity Centre Switzerland (QA/SAC-CH) which is supported by MeteoSwiss and Empa. OBP thanks to Russian Fond for Basic Research (RFBR), grant No. 20-55-12001. We acknowledge the use of data and imagery from LANCE FIRMS operated by NASA's Earth Science Data and Information System (ESDIS) with funding provided by NASA Headquarters.

**Author contributions**

Experimental work at the sampling site was done by DLN. DLN performed IDTD-GC-TOF-MS and thermal-optical carbon analyses with assistance from JO and GA. DLN, HC and JSK analyzed IDTD-GC-TOF-MS and TOR data; HC conducted the statistical analysis of IDTD-GC-TOFMS data; NAN, NB and MS were the principal investigators of the long-term aerosol and trace gas observations; MS and DLN performed the trace gas data analyses; SH conducted the back trajectory analysis. Data
interpretation was done by DLN, HC, SMP, MS, SH, OBP and JSK. HC, JSK, GE, RZ and XAN supervised the study. XAN, DLN, OBP and JSK acquired the funding. The manuscript was written by DLN, HC and SMP with contribution from all authors.

**Data and code availability**

The continuous trace gas records from Pha Din are available on the dedicated GAW data repositories, i.e. $O_3$ data can be found
at the World Data Centre for Reactive Gases ([www.gaw-wdcrg.org](http://www.gaw-wdcrg.org)), while $CO_2$, $CH_4$, and CO data are available on the World Data Centre for Greenhouse Gases ([https://gaw.kishou.go.jp/](https://gaw.kishou.go.jp/)). The trajectory model FLEXTRA is open source software that can be obtained from [www.flexpart.eu](http://www.flexpart.eu). Trajectory plots for the site PDI can be inspected at the following link: [https://lagrange.empa.ch/FLEXTRA_browser/data_access.php?stat=PDI&year=2015&prod=_hourly_](https://lagrange.empa.ch/FLEXTRA_browser/data_access.php?stat=PDI&year=2015&prod=_hourly_). Further data is available from the corresponding author upon request.

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
