# Peer review of "Carbonaceous aerosol composition in air masses influenced by largescale biomass burning: a case-study in Northwestern Vietnam"

_Atmospheric Chemistry and Physics, 2020_

## Referee Comment (RC1) · Anonymous Referee #1 · 20 Dec 2020

This authors report ground-based measurements of speciated OA, OC, EC, and various gas-phase components in northern Vietnam. The measurements are divided into three categories, including low, medium, and high biomass burning periods based on different tracers (including levoglucosan). The manuscript is well written, and the data analysis provides a reasonable way to source apportionment the sampled emissions. The manuscript merits publication in ACP after the authors address the below comments. I have two main concerns with the manuscript.

1) The authors do not really emphasize the importance of the work. My impression after reading the manuscript was that the main motivation for the work was the lack of measurements in the area, which significantly weakens the quality of the measurements reported and the effort that was put in the analysis of the data. The results presented are not placed in a greater context. For instance, how are such measurements useful to the community? What did the authors learn? This seems largely missing.

2) The section reporting the molecular composition of aerosol constituents (section 3.2.2) is very linear and, at times, doesn't converge to any meaningful conclusions. For instance, in the first paragraph of that section (anhydrosugars), the only useful information is reported concentrations. The rest of the discussion is not relevant to the overall message of the paper and seemed to belong to methods and not results. The same is true of the PAH discussion. Here, the authors discuss in detail the trends of various indicators (e.g., BaP/BeP+BaP) but the main take away from that section is not clear to me. To illustrate this, the authors mention the following: "Altogether, PAH diagnostic ratios and PAH pattern are not useful in this data set to elucidate their emission sources and could not linked to BB, which agrees with the relatively large distance of PAH to BB markers such as anhydrous sugars and methoxyphenols". Why is this interesting? Is it because other studies have used these ratios for source apportionment, and that these diagnostics are not valid here? If so, this needs to me clearly motivated and better discussed. If not, then this should be omitted from the result section. In summary, section 3.2.2. needs to be tightened and focused.

Minor comments:

Line 21: "but OC composition studies are missing in the scientific literature." Change to: "but OC composition studies from this site are missing in the scientific literature".

Line 53: "...may change (Nordin et al., 2015; Gilmour et al., 2015)." This is disconnected from the sentence. Please adjust.

Line 65: define PM2.5

Line 67: define PAH and o-PAH

Line 75: adjust to: "Aside from the measurements..."

Line 79: 579,000 and not 579'000

Line 82: adjust to: "in the vicinity of station."

Line 86: adjust to: (fine inhalable particles <2.5 um diameter)"

Line 97: remove "from" after Pyrolytic OC (OP)

Line 173: replace "distinctly different above one" with "larger than unity"

Line 175: add space between EC1apparent and OP for clarity

Line 204: the authors offer no explanation as to why nitrophenols behave so distinctly from the rest of the categories. Is it because nitrophenols are secondary and the rest are primary emissions? This needs to be expanded.

Line 217: replace "quantified OA" with "speciated OA"

Section 3.2.2. see major comment #2

Line 236: adjust to" appeared in the range of"

Line 255: replace "dominating" with "significant"

Figure 6: the arrows in the graphs are confusing and I am not sure what the authors are trying to point at

Line 376: the differences in behavior for CO, CO2, CH4 and O3 is also due to the fact that CO, CO2, CH4 are mainly of primary origin whereas O3 is not.

Line 401: ", and rose again..."

Line 404: "As for the $\Delta O3/\Delta CO$, both CO and CO2 are close to the chosen background. Consequently, at CO2 mixing ratios below 410 ppm, the uncertainty of MCE exceeds 0.04.". These sentences seem unrelated to me, and the first sentence makes no sense.

Line 407: I am surprised that Figure S5 is buried in the SI. To me, Figure S5 is a central finding of the paper. The manuscript first discusses timeseries of multiple speciated OA components and their variability. The authors then discuss the variability in terms of three emission categories (low, med, and high BB), but MCE seems like a more intuitive metric. Also, MCE is easily obtained from emission inventories and the parameterizations presented in Figure S5 can be useful to modelers. MCE explains part of the measured variability for primary components, as already noted in the text, and (to a lesser extent) for secondary components. The results are robust given that multiple sources influence the sampled air.

Line 407: "The absolute values of the slopes are similar to the intercepts obtain from the regression (Table S1), giving reasonable concentrations of organic close to detection of ideal combustion with MCE of 1.". This sentence is not clear to me. I think what the authors are trying to say is the following: to have null concentrations of OA at MCE = 1 (perfect combustion), the slope of the regression should be equal to minus the intercept. I get that. Is this right? Anyway, please re-phrase.

Line 409: "PAHs and -alkanes, which cannot be formed by atmospheric aging, show the best agreement with the regression function, whereas the fit for nitrophenols is clearly dominated by the datapoints of the high-BB days, indicating formation by secondary processes.". I have trouble with this. If I use the same rational the authors previously used, if nitrophenols (for instance) are likely SOA, then I would not expect a value $\sim$0 near MCE = 1 (since it is not a primary combustion pollutant and therefore not directly driven by changes in MCE). However, the values of the slope and intercept for nitrophenols are very close (see Table S1). Wouldn't you expect in this case a significant difference between the slope and y-intercept of the regression line (so as to not get a value of 0 at MCE=1)? I understand that there is substantial variability in the measurements, but the fit is consistent with a value of near zero at an MCE of unity, suggesting the same behavior as combustion derived POA. Please explain. Also note that there are 2 tables in the SI named Table S1. Adjust in the SI and in the manuscript.

Line 426: "A main difference between the more polluted cluster (high) and the medium polluted cluster (medium) seems to be a recirculation over land area for the medium cluster as compared to a more westerly advection for the high cluster." This also explains the lower $\Delta O3/\Delta CO$ ratio for high BB event compared to the medium BB event.

Line 426: The significantly higher concentrations of nitrophenols (presumably SOA) during the high BB event compared to the medium BB event seems in disagreement with the fact the high BB plume is fresher. Clarify.

Line 443: replace "comprehensive" with "extensive"

---

## Referee Comment (RC2) · Anonymous Referee #2 · 29 Dec 2020

General comments

The manuscript is generally well-written and straightforward to follow. It documents greater details of OA composition in "a region of scarce data availability", so would be of value to extend available literature on BB OA in this region. After some specific comments and many technical corrections are made, the manuscript would be suitable for publication. In particular, there are lots of technical errors with figure captions and style. A major though technical suggestion would be to revise clustergram color scheme and its color scheme referenced in subsequent figures so it is red-green color blind accessible.

Specific comments
1. Clustering analysis color scheme used throughout figures: The use of green and red for ⅔ clusters throughout the figures without marker type changes would be confusing for any reader with red-green color blindness. Would recommend revising the selected color scheme throughout figures and text and/or implement line type and marker type changes in figures to aid.
2. Line 270: Rephrase the part following "...volatilization," as I think authors are trying to say that absorption leads to these compounds becoming trapped into the particle phase rather than emitted via absorption
3. Line 341: rephrase to "...nitrophenols can rapidly form from monoaromatics photoxidation or their reactions with nitrate radicals."
4. Line 342: rephrase sentence for clarity
5. Line 344: why do the authors suggest nighttime transport here given that they state 4-NC formation is fast at night? Is there something about the back trajectories that support formation of 4-NC away from site and transported at nighttime vs daytime to sight? Seems too speculative.
6. Introduction and Conclusions sections: The manuscript could provide greater appreciation to readership by additional text providing recommendations on what further data/measurements are needed in this area going forward and what larger scale atmospheric problems are to be addressed here. How does BB at PDI differ from other areas impacted by high BB influence referenced throughout (tropical forests, etc).

Technical corrections
1. Line 51: consider rephrasing sentence; unclear
2. Line 79: change quotation to comma in population number
3. Line 180: rephrase (1) insert "than" after "less" and (2) delete "of"
4. Line 247: delete "and" before "... the most abundant…"
5. Line 273: delete "were" before "...ranged from..."
6. Line 297: insert "do" after "nor"
7. Line 322: correct spelling of "concertation"
8. Line 333: change "are" to "of" or rephrase sentence
9. Line 347: fix figure reference to figure 6
10. Line 445: insert "respectively, " following the concentrations of OC and EC

11. Figure 2 not referenced in main text
12. Figure 3a right axis units should be ng/m3
13. Figure 3b) caption: language in caption regarding ratio of char-EC to soot-EC reversed from figure including description of dashed lines
14. Figure 5 caption: says left axes in units of ug/m3, but some axes show ng/m3 levels
15. Figure 7 caption: fix cross reference to section on clustergram analysis; not Section 3.3

---

## Author Comment (AC1) · 18 Feb 2021

We thank both referees for their comments and provide here a point-by-point reply to the remarks submitted by anonymous referee 1 (in black font) in this document. Our reply is provided in blue font. Changes in the manuscript are available through the attached track-change version.

**Anonymous Referee #1:** The authors report ground-based measurements of speciated OA, OC, EC, and various gas-phase components in northern Vietnam. The measurements are divided into three categories, including low, medium, and high biomass burning periods based on different tracers (including levoglucosan). The manuscript is well written, and the data analysis provides a reasonable way to source apportionment the sampled emissions. The manuscript merits publication in ACP after the authors address the below comments. I have two main concerns with the manuscript.

**Reply:** We thank the referee for their availability and the generally positive feedback. The detailed comments are addressed below.

1) The authors do not really emphasize the importance of the work. My impression after reading the manuscript was that the main motivation for the work was the lack of measurements in the area, which significantly weakens the quality of the measurements reported and the effort that was put in the analysis of the data. The results presented are not placed in a greater context. For instance, how are such measurements useful to the community? What did the authors learn? This seems largely missing.

**Reply:** Indeed, filling data gaps in data scarce area is one of the main motivators for measurements at the Pha Din (PDI) regional GAW station. The availability of reliable scientific data and information on the chemical composition of the atmosphere is crucial for a sound assessment of air quality pollution sources and impacts of future global change. To get a full coverage such data must be consistent, of adequate quality, and have to be available from various locations world-wide. Spatial data coverage considerably improved in recent years, also through the initiation to establish additional GAW stations in locations such as PDI, Vietnam, aside of other stations in South-East Asia, Central Asia, Africa and South America. However, besides the initial work to get those measurements running, further efforts are needed to ensure a sustainable effect and foster scientific use of the data. Data from PDI are particularly useful to study recurrent large-scale biomass burning (BB) on the Indochinese Peninsula. BB is a globally widespread phenomenon, and emissions characterization of high scientific and societal relevance. The fires release pollutants, which are harmful for human and ecosystem health and alter the Earth's radiative balance. Yet, the impact of various types of BB on the global radiative forcing remains poorly constrained concerning greenhouse gas emissions, BB organic aerosol (OA) chemical composition and related light absorbing properties. Fire emissions composition is influenced by multiple factors (e.g., fuel and thereby vegetation-type, fuel moisture, fire temperature, available oxygen). Due to regional variations in these parameters, studies in different world regions are needed. PDI is well suited to study the large-scale fires on the Indochinese Peninsula, whose pollution plumes are frequently transported towards the site, and, because other local pollution sources are comparatively low at PDI, can be studied almost undisturbed. We exploit the GHG data available within GAW and add further detailed analysis of chemical carbonaceous aerosol composition from a 3-weeks intensive campaign in our first study; in future work, longer-term measurements and additional parameters might be investigated, and altogether the data give insight on health- and climate-change related properties and quantities from BB on the Indochinese Peninsula. We have revised the manuscript abstract, introduction and conclusions in order to transfer this message and the associated relevance of our study better to the readers. Please see the track change version.

2) The section reporting the molecular composition of aerosol constituents (section 3.2.2) is very linear and, at times, doesn't converge to any meaningful conclusions. For instance, in the first paragraph of that section (anhydrosugars), the only useful information is reported concentrations. The rest of the discussion is not relevant to the overall message of the paper and seemed to belong to methods and not results. The same is true of the PAH discussion. Here, the authors discuss in detail

the trends of various indicators (e.g., BaP/BeP+BaP) but the main take away from that section is not clear to me. To illustrate this, the authors mention the following: "Altogether, PAH diagnostic ratios and PAH pattern are not useful in this data set to elucidate their emission sources and could not linked to BB, which agrees with the relatively large distance of PAH to BB markers such as anhydrous sugars and methoxyphenols". Why is this interesting? Is it because other studies have used these ratios for source apportionment, and that these diagnostics are not valid here? If so, this needs to me clearly motivated and better discussed. If not, then this should be omitted from the result section. In summary, section 3.2.2. needs to be tightened and focused.

**Reply:** We intended a point-by-point discussion of compound classes, yet, agree, that section 3.2.3. lacks concluding statements. In order to support conclusions about the contribution of BB and atmospheric aging to OA, Figure 5 was extended by simple source apportionment of OC with relative contributions of OC from BB ($OC_{BB}$) and secondary formation ($OC_{sec}$). Furthermore, we added information on the relation between compound classes and anhydrosugars in order to investigate if molecular markers stem from the same BB source over the entire sampling period during the manuscript revision; speciation of individual anhydrosugars was obtained by data published in Bukowiecki et al. (2019) (s. additional figures and table in supplement: Figures S3, S7, S8 and S9; Table S2). From these additional molecular marker ratios, we could specify further the type of BB.

Specific remarks:

- The compound class of "anhydrosugars" consists of levoglucosan, mannosan and galactosan, of which levoglucosan is an unambiguous marker for biomass burning (BB). More details about the BB source may be derived from its ratios to the other two components, but because of overlapping peaks in GC analysis, the results are highly uncertain thus not presented in our initial manuscript. For our revision, we resorted to the data from a different analytical measurement system published in our companion work by Bukowiecki et al. (2019). We are now able to discuss possible BB sources based on the ratios of levoglucosan to mannosan in our revision.
- The section on methoxyphenols was extended by applying diagnostic ratios of syringic acid and vanillic acid for BB source identification and syringic acid to syringaldehyde for plume age determination.
- PAH diagnostic ratios are frequently used as indication of dominating emission sources (e.g. Tobiszewski & Namiesnik 2012; Galarneau 2008; Ravindra et al 2008), so we applied compiled PAH diagnostic ratios from reviews on our data set describing large influence of a single emission source biomass burning. "Publication bias", meaning a preferred publication of positive results, may alter the impression of the performance of proposed parametrization. Therefore, despite a missing distinct outcome, we decided that this is an outcome as well and deserves to be presented with additional possible reasons for this disagreement (atmospheric aging, gas-particle-partitioning). To give the negative result less weight in the manuscript, the figure according to PAH diagnostic ratios was revised and put in the supplemental material (Figure S6).

Minor comments:

Line 21: "but OC composition studies are missing in the scientific literature." Change to: "but OC composition studies from this site are missing in the scientific literature".

**Reply:** "from this site" was added.

Line 53: "...may change (Nordin et al., 2015; Gilmour et al., 2015)." This is disconnected from the sentence. Please adjust.

**Reply:** The sentence was reorganized and an incorrect citation (Gilmour et al. 2015) was replaced (by Li et al. 2021).

Line 65: define PM2.5

**Reply:** PM2.5 was defined as "atmospheric fine particulate matter with aerodynamic diameter $\leq 2.5$ µm".

Line 67: define PAH and o-PAH

**Reply:** Done.

Line 75: adjust to: "Aside from the measurements..."

**Reply:** Done.

Line 79: 579,000 and not 579'000

**Reply:** Changed.

Line 82: adjust to: "in the vicinity of station."

**Reply:** Done.

Line 86: adjust to: (fine inhalable particles <2.5 um diameter)"

**Reply:** Done.

Line 97: remove "from" after Pyrolytic OC (OP)

**Reply:** Removed.

Line 173: replace "distinctly different above one" with "larger than unity"

**Reply:** Replaced.

Line 175: add space between EC1apparent and OP for clarity

**Reply:** Done.

Line 204: the authors offer no explanation as to why nitrophenols behave so distinctly from the rest of the categories. Is it because nitrophenols are secondary and the rest are primary emissions? This needs to be expanded.

**Reply:** We added that nitrophenols may be formed during atmospheric oxidation of e.g., phenols and benzenediols in the presence of $NO_x$ during few days of atmospheric aging. This topic is more in detail discussed with respect to the three individually analyzed nitrophenols in section 3.2.3.7 with recent findings on nitrophenol origins in East Asia from Salvador et al. (2021).

Line 217: replace "quantified OA" with "speciated OA"

**Reply:** Corrected.

Section 3.2.2. see major comment #2

**Reply:** We assume the referee refers to section 3.2.3. (OA speciation) rather than the description of the clustergram result in section 3.2.2. We extended the discussion of molecular markers based on diagnostic ratios and correlations to anhydrosugars for convergence into conclusions.

Line 236: adjust to" appeared in the range of"

**Reply:** "the" was added.

Line 255: replace "dominating" with "significant"

**Reply:** Done.

Figure 6: the arrows in the graphs are confusing and I am not sure what the authors are trying to point at

**Reply:** We added information in the caption that the gray-shaded indicates higher contribution of pyrogenic PAH and more intense atmospheric aging and removed the arrows.

Line 376: the differences in behavior for CO, CO2, CH4 and O3 is also due to the fact that CO, CO2, CH4 are mainly of primary origin whereas O3 is not.

**Reply:** We agree with the referee and revised the phrases in line 376-378

Line 401: ", and rose again..."

**Reply:** Done.

Line 404: "As for the ΔO3/ΔCO, both CO and CO2 are close to the chosen background. Consequently, at CO2 mixing ratios below 410 ppm, the uncertainty of MCE exceeds 0.04.". These sentences seem unrelated to me, and the first sentence makes no sense.

**Reply:** We deleted the first part of sentence one and merged the two sentences.

Line 407: I am surprised that Figure S5 is buried in the SI. To me, Figure S5 is a central finding of the paper. The manuscript first discusses timeseries of multiple speciated OA components and their variability. The authors then discuss the variability in terms of three emission categories (low, med, and high BB), but MCE seems like a more intuitive metric. Also, MCE is easily obtained from emission inventories and the parameterizations presented in Figure S5 can be useful to modelers. MCE explains part of the measured variability for primary components, as already noted in the text,

and (to a lesser extent) for secondary components. The results are robust given that multiple sources influence the sampled air.

**Reply:** We thank the referee for this comment and moved Figure S5 to the main text. We have added a new section "Parametrization of Organic Aerosol constituents with MCE" with a more detailed discussion of this figure.

Line 407: "The absolute values of the slopes are similar to the intercepts obtain from the regression (Table S1), giving reasonable concentrations of organic close to detection of ideal combustion with MCE of 1.". This sentence is not clear to me. I think what the authors are trying to say is the following: to have null concentrations of OA at MCE = 1 (perfect combustion), the slope of the regression should be equal to minus the intercept. I get that. Is this right? Anyway, please re-phrase.

**Reply:** We rephrased the sentence accordingly.

Line 409: "PAHs and -alkanes, which cannot be formed by atmospheric aging, show the best agreement with the regression function, whereas the fit for nitrophenols is clearly dominated by the datapoints of the high-BB days, indicating formation by secondary processes.". I have trouble with this. If I use the same rational the authors previously used, if nitrophenols (for instance) are likely SOA, then I would not expect a value ~0 near MCE = 1 (since it is not a primary combustion pollutant and therefore not directly driven by changes in MCE). However, the values of the slope and intercept for nitrophenols are very close (see Table S1). Wouldn't you expect in this case a significant difference between the slope and y-intercept of the regression line (so as to not get a value of 0 at MCE=1)? I understand that there is substantial variability in the measurements, but the fit is consistent with a value of near zero at an MCE of unity, suggesting the same behavior as combustion derived POA. Please explain. Also note that there are 2 tables in the SI named Table S1. Adjust in the SI and in the manuscript.

**Reply:** In our opinion, a value close to 0 at MCE of 1 does not contradict the conclusion about the possible secondary origin of nitrophenols. Our data indicate that the BB plumes is relatively fresh (may have undergone only short atmospheric aging). Nitrophenols may be primary plume constituents, but also formed from atmospheric oxidation of phenol or phenol-derivatives at high-NOx conditions (Kroflic et al., 2015). For 4-nitrophenol, peaking concentrations at low equivalent photochemical ages have been observed in laboratory-aging of wood combustion aerosol (Hartikainen et al., 2020). We added this information to the discussion of the relation between OA constituents and MCE.

Line 426: "A main difference between the more polluted cluster (high) and the medium polluted cluster (medium) seems to be a recirculation over land area for the medium cluster as compared to a more westerly advection for the high cluster." This also explains the lower ΔO3/ΔCO ratio for high BB event compared to the medium BB event.

**Reply:** We thank the referee for this advice. We included the recirculation as possible reason for lower ΔO3/ΔCO , but also expanded the discussion on ΔO3/ΔCO by also considering background NO$_x$ on ozone formation.

Line 426: The significantly higher concentrations of nitrophenols (presumably SOA) comment during the high BB event compared to the medium BB event seems in disagreement with the fact the high BB plume is fresher. Clarify.

**Reply:** We have commented on the nitrophenol aspect in our reply to two other comments by the referee 1 earlier. Because nitrophenols may, aside of being primary plume constituents, derive as early or first-generation oxidation products with peak concentrations after comparably short atmospheric aging and declining toward higher equivalent photochemical ages as shown for laboratory-aging of wood combustion emissions (Hartikainen et al., 2020). Therefore, the observation of higher nitrophenol concentration during high-BB air is in line with the estimated higher age of the aerosol of medium-BB days compared to high-BB days. We added some discussion about that topic and one additional very recent reference by Salvador et al. (2021) in the section on parametrization of OA constituents and MCE.

Line 443: replace "comprehensive" with "extensive"

**Reply:** Done.

**References**

Bukowiecki, N., Steinbacher, M., Henne, S., Nguyen, N. A., Nguyen, X. A., Le Hoang, A., Nguyen, D. L., Duong, H. L., Engling, G., Wehrle, G., Gysel-Beer, M., and Baltensperger, U.: Effect of Large-scale Biomass Burning on Aerosol Optical Properties at the GAW Regional Station Pha Din, Vietnam, Aerosol Air Qual. Res., 19, 1172–1187, https://doi.org/10.4209/aaqr.2018.11.0406, 2019.

Galarneau, E.: Source specificity and atmospheric processing of airborne PAHs: Implications for source apportionment, Atmos. Environ., 42, 8139–8149, https://doi.org/10.1016/j.atmosenv.2008.07.025, 2008.

Hartikainen, A., Tiitta, P., Ihalainen, M., Yli-Pirilä, P., Orasche, J., Czech, H., Kortelainen, M., Lamberg, H., Suhonen, H., Koponen, H., Hao, L., Zimmermann, R., Jokiniemi, J., Tissari, J., and Sippula, O.: Photochemical transformation of residential wood combustion emissions: dependence of organic aerosol composition on OH exposure, Atmos. Chem. Phys., 20, 6357–6378, https://doi.org/10.5194/acp-20-6357-2020, 2020.

Kroflič, A., Grilc, M., and Grgić, I.: Does toxicity of aromatic pollutants increase under remote atmospheric conditions?, Sci. Rep., 5, 8859, https://doi.org/10.1038/srep08859, 2015.

Ravindra, K., Sokhi, R., and van Grieken, R.: Atmospheric polycyclic aromatic hydrocarbons: Source attribution, emission factors and regulation, Atmos. Environ., 42, 2895–2921, https://doi.org/10.1016/j.atmosenv.2007.12.010, 2008.

Salvador, C. M. G., Tang, R., Priestley, M., Li, L., Tsiligiannis, E., Le Breton, M., Zhu, W., Zeng, L., Wang, H., Yu, Y., Hu, M., Guo, S., and Hallquist, M.: Ambient nitro-aromatic compounds – biomass burning versus secondary formation in rural China, Atmos. Chem. Phys., 21, 1389–1406, https://doi.org/10.5194/acp-21-1389-2021, 2021.

Tobiszewski, M. and Namieśnik, J.: PAH diagnostic ratios for the identification of pollution emission sources, Einviron. Pollut., 162, 110–119, https://doi.org/10.1016/j.envpol.2011.10.025, 2012.

---

## Author Comment (AC2) · 18 Feb 2021

We thank both referees for their comments and provide here a point-by-point reply to the remarks submitted by anonymous referee 2 (in black font) in this document. Our reply is provided in blue font. Changes in the manuscript are available through the attached track-change version.

**Anonymous Referee #2:** General comments: The manuscript is generally well-written and straightforward to follow. It documents greater details of OA composition in "a region of scarce data availability", so would be of value to extend available literature on BB OA in this region. After some specific comments and many technical corrections are made, the manuscript would be suitable for publication. In particular, there are lots of technical errors with figure captions and style. A major though technical suggestion would be to revise clustergram color scheme and its color scheme referenced in subsequent figures so it is red-green color blind accessible.

**Reply:** We thank the referee for their availability and the generally positive feedback. We have corrected the technical errors and revised the color scheme of the separation of low-, medium and high-BB days. Specific comments are addressed below.

Specific comments

1.      Clustering analysis color scheme used throughout figures: The use of green and red for ⅔ clusters throughout the figures without marker type changes would be confusing for any reader with red-green color blindness. Would recommend revising the selected color scheme throughout figures and text and/or implement line type and marker type changes in figures to aid.

**Reply:** We changed the color scheme highlighting the different periods and added different symbols when applicable.

2.      Line 270: Rephrase the part following "...volatilization," as I think authors are trying to say that absorption leads to these compounds becoming trapped into the particle phase rather than emitted via absorption

**Reply:** Yes, the referee is right. We rephrased the sentence to clarify.

3.      Line 341: rephrase to "...nitrophenols can rapidly form from monoaromatics photoxidation or their reactions with nitrate radicals."

**Reply:** We have revised the discussion around nitrophenols further based on comments from referee 1.

4.      Line 342: rephrase sentence for clarity

**Reply:** Done.

5.      Line 344: why do the authors suggest nighttime transport here given that they state 4-NC formation is fast at night? Is there something about the back trajectories that support formation of 4-NC away from site and transported at nighttime vs daytime to sight? Seems too speculative.

**Reply:** We thank the referee for the comment. Generally, nitrophenols may be of primary origin, or formed as secondary compounds during atmospheric aging (see, for instance, also a recent article by Salvador et al., 2021). The atmospheric aging might be initiated during day-time (primarily through OH radicals) or at night-time (primarily through NO3 radicals). However, their formation depends on the availability of high NOx levels (in particular NO2). Owing the photo-labile nature of NO2 as well as the nitrophenols themselves, we suggested that nitrophenols during dark periods (night) might be the more prominent way of nitrophenol formation in our samples. However, as the samples

integrate over 24 hours, we cannot decisively conclude on the prominent pathway. We have revised the discussion around nitrophenols also based on comments from referee 1; please see the track-changed version.

6.    Introduction and Conclusions sections: The manuscript could provide greater appreciation to readership by additional text providing recommendations on what further data/measurements are needed in this area going forward and what larger scale atmospheric problems are to be addressed here. How does BB at PDI differ from other areas impacted by high BB influence referenced throughout (tropical forests, etc).

**Reply:** We thank the referee for this valuable comment. We have revised Abstract, Introduction and Conclusions to address the wider impact of our results. As also highlighted in our reply to referee 1, data from PDI are particularly useful to study recurrent large-scale biomass burning (BB) on the Indochinese Peninsula. BB is a globally widespread phenomenon, and emissions characterization of high scientific and societal relevance. The fires release pollutants, which are harmful for human and ecosystem health and alter the Earth's radiative balance. Yet, the impact of various types of BB on the global radiative forcing remains poorly constrained concerning greenhouse gas emissions, BB organic aerosol (OA) chemical composition and related light absorbing properties. Fire emissions composition is influenced by multiple factors (e.g., fuel and thereby vegetation-type, fuel moisture, fire temperature, available oxygen). Due to regional variations in these parameters, studies in different world regions are needed. PDI is well suited to study the large-scale fires on the Indochinese Peninsula, whose pollution plumes are frequently transported towards the site, and, because other urban pollution is comparatively low, can be studied almost undisturbed. Please have a look at the track-change version for the updated abstract, introduction and conclusions.

Technical corrections

1.    Line 51: consider rephrasing sentence; unclear

**Reply:** Done.

2.    Line 79: change quotation to comma in population number

**Reply:** Done.

3.    Line 180: rephrase (1) insert "than" after "less" and (2) delete "of"

**Reply:** Done.

4.    Line 247: delete "and" before "... the most abundant…"

**Reply:** Done.

5.    Line 273: delete "were" before "...ranged from..."

**Reply:** Done.

6.    Line 297: insert "do" after "nor"

**Reply:** Done.

7.    Line 322: correct spelling of "concertation"

**Reply:** "Concertation" was changed to "concentration".

8.    Line 333: change "are" to "of" or rephrase sentence

**Reply:** Done.

9.    Line 347: fix figure reference to figure 6

**Reply:** Done.

10.    Line 445: insert "respectively, " following the concentrations of OC and EC

**Reply:** Done.

11.    Figure 2 not referenced in main text

**Reply:** Figure 2 was referenced in line 145 of the article.

12.    Figure 3a right axis units should be ng/m3

**Reply:** Done.

13.    Figure 3b) caption: language in caption regarding ratio of char-EC to soot-EC reversed from figure including description of dashed lines

**Reply:** We changed the order of ratios inside the figure.

Figure 5 caption: says left axes in units of ug/m3, but some axes show ng/m3 levels

**Reply:** The caption was changed to ng m$^{-3}$.

14.    Figure 7 caption: fix cross reference to section on clustergram analysis; not Section 3.3

**Reply:** Done.

**References**

Salvador, C. M. G., Tang, R., Priestley, M., Li, L., Tsiligiannis, E., Le Breton, M., Zhu, W., Zeng, L., Wang, H., Yu, Y., Hu, M., Guo, S., and Hallquist, M.: Ambient nitro-aromatic compounds – biomass burning versus secondary formation in rural China, Atmos. Chem. Phys., 21, 1389–1406, https://doi.org/10.5194/acp-21-1389-2021, 2021.

---

## Author Response (AR3)

Dear Dr. Dubey,

We would like to resubmit our manuscript "*Carbonaceous aerosol composition in air masses influenced by large-scale biomass burning: a case-study in Northwestern Vietnam*" for your consideration in the Special Issue *The role of fire in the Earth system: understanding interactions with the land, atmosphere, and society.* We have included the two suggested publications on brown carbon on line 428 and following and look forward to your decision.

Sincerely,

Hendryk Czech
* * *
Dr. Hendryk Czech

University of Rostock
Department of Analytical and Technical Chemistry
Dr.-Lorenz-Weg 2
18059 Rostock (Germany)
phone: (+49) 381 / 498 6533
email: hendryk.czech@uni-rostock.de

Helmholtz Zentrum München
German Research Centre for Environmental Health (GmbH)
Cooperation Group "Comprehensive Molecular Analytics" (CMA)
Gmunder Straße 37
81379 München (Germany)